



# Responses of an abyssal meiobenthic community to short-term burial with crushed nodule particles in the South-East Pacific

Lisa Mevenkamp[1], Katja Guilini[1], Antje Boetius[2], Johan De Grave[3], Brecht Laforce[4], Dimitri Vandenberghe[3], Laszlo Vincze[4], Ann Vanreusel[1]

5 [1] Department of Biology, Marine Biology Research Group, Ghent University, Ghent, Belgium
[2] HGF MPG Joint Research Group for Deep-Sea Ecology and Technology, Max Planck Institute for Marine Microbiology, Celsiusstr. 1, Bremen, Germany
[3] Department of Geology, Mineralogy and Petrology Research Unit, Ghent University, Ghent, Belgium
[4] Department of Chemistry, X-ray Imaging and Microspectroscopy Research Group, Ghent University, Ghent, Belgium

10 *Correspondence to*: Lisa Mevenkamp (Lisa.Mevenkamp@ugent.be)

**Abstract.** Increasing industrial metal demands due to rapid technological developments may drive the prospection and exploration of deep-sea mineral resources such as polymetallic nodules. To date, the potential environmental consequences of mining operations in the remote deep sea are poorly known. Experimental studies are scarce, especially with regard to the effect of sediment and nodule debris depositions as a consequence of seabed mining. To elucidate the potential effects of the 15 deposition of crushed polymetallic nodule particles on abyssal meiobenthos communities, a short (11 days) *in situ* experiment at the Peru Basin in the South East Pacific Ocean was conducted. We covered abyssal, soft sediment with approx. 2 cm of crushed nodule particles and sampled the sediment after eleven days of incubation at 4200 m water depth. Short-term ecological effects on the meiobenthos community were studied including changes in their composition and vertical distribution in the sediment as well as nematode genus composition. Additionally, copper burden in a few similar-sized, but randomly selected 20 nematodes was measured by means of μ-X-ray fluorescence. At the end of the experiment, $46 \pm 1$ % of the total meiobenthos occurred in the added crushed nodule layer while abundances decreased in the underlying 2 cm compared to the same depth-interval in original, undisturbed sediments. Densities and community composition in the deeper 2-5 cm layers remained similar in covered and undisturbed sediments. The migratory response into the added substrate was particularly seen in polychaetes ($73 \pm 14$ %, relative abundance across all depth layers) copepods ($71 \pm 6$ %), nauplii ($61 \pm 9$ %) and nematodes ($43 \pm 1$ %). 25 While the dominant nematode genera in the added substrate did not differ from those in underlying layers or the undisturbed sediments, feeding type proportions in this layer were altered with a 9 % decrease of non-selective deposit feeders and an 8 % increase in epistrate feeders. Nematode tissue copper burden did not show elevated copper toxicity resulting from burial with crushed nodule particles. The results indicate that short-term substrate burial requires special attention with regard to ecological consequences of mineral extraction in the deep-sea.



# 1    Introduction

The interest in mineral deposits from the deep seafloor commenced in the early 70s, after the discovery of a widespread occurrence of economically valuable polymetallic nodules (Glasby, 2000; Mero, 1977). However, economic and technological limitations of using deep-sea resources at that time hampered further industrial activities. The advancements in deep-sea technology and other socio-economic developments have led to a new surge for deep-sea minerals in the past decade and legal frameworks are being developed to manage their extraction in international waters (Lodge et al., 2014). Polymetallic nodules are decimetres-size concretions of ore lying on the surface of abyssal sediments in 4000 – 6500 m water depth and cover large areas of the Pacific and Indian Ocean (Hein and Koschinsky, 2014). Besides the high content in valuable metals such as copper, nickel and cobalt, nodules exhibit a high porosity, low bulk density and fine grain size with very slow formation and growth rates of less than 250 mm My$^{-1}$ (million years) (Jain et al., 1999; Von Stackelberg, 2000). These properties result in very brittle structures that are easily damaged or broken when applying low force (Charewicz et al., 2001; Jain et al., 1999; Thiel et al., 1993; Zenhorst, 2016). Therefore, breakage and abrasion of nodule particles is likely to occur during a mining operation with heavy gear.

Polymetallic nodule mining is hence expected to have various direct and indirect environmental impacts such as nodule removal, removal of surface sediment, sediment compaction, sediment suspension and deposition, discharge of waste material and potential release of toxic amounts of heavy metals (Clark and Smith, 2013; Rolinski et al., 2001; Sharma et al., 2001; Thiel, 2001). Additionally, nodule particles abraded during collection may get mixed with the suspended sediment and redeposited in areas close to or further away of the mined site, depending on their sedimentation rate. An economically viable mining operation would cover an area of 300-800 km² per year (Smith et al., 2008) and after 20 years an estimated 8500 km² would have been mined per concession area (Madureira et al., 2016). Such a large-scale mining operation is expected to directly impact the nodule associated fauna (Purser et al., 2016; Vanreusel et al., 2016). However, deposition of sediment and nodule particles on the seafloor resulting from mining activities may also impact the typical abyssal soft-sediment fauna, but knowledge about the direct responses of those organisms to substrate deposition is scarce.

In the abyssal deep sea, the meiobenthos (32 - 1000 µm) constitute the most dominant metazoan component of infaunal communities in terms of biomass (Rex et al., 2006). Due to their high abundance, meiofauna play an important role for the energy flow inside abyssal sediments but also for the functioning of the infaunal ecosystem through e.g. bioturbation, degradation of organic matter or species interactions (Schratzberger and Ingels, 2017). Moreover, meiofauna contribute greatly to the high biodiversity of abyssal ecosystems with nematodes in particular being the most diverse metazoan taxon in some deep-sea habitats (Sinniger et al., 2016). Typically, meiofaunal generation times vary in the range of weeks to months, depending on the species (Coull, 1999; Gerlach, 1971). However, this has only been assessed for shallow water species so far and generation times may be longer in the deep sea where many taxa are characterize by a high longevity (Cailliet et al., 2001; Giere, 2009).



Due to their residence inside the sediment, nodule mining will inevitably disturb meiobenthic communities, directly or indirectly. Directly through the removal of the sediment surface layers, which causes removal and redistribution of meiobenthic organisms, and indirectly through sediment deposition, which may have consequences for the survival and vertical structuring of underlying meiobenthic communities.

Previous research on the direct effect of nodule mining suggests that abyssal benthic communities have the capacity to recover from small scale sediment disturbances (Gollner et al., 2017), although effects of stress by pollution, oxygen deficiency, forced migration etc. on overall fitness has not been investigated. In that respect, full recovery of a disturbed community may be a long lasting process which may still be incomplete several decades after the disturbance (Gollner et al., 2017). Previous findings are based on small-scale disturbance scenarios where nodules were removed or ploughed (overview given in Jones et

al., 2017). In general, recovery of mobile fauna occurred faster than that of sessile fauna and small organisms tend to recover faster than large organisms (Gollner et al., 2017; Jones et al., 2017). These deep-sea experiments clearly indicated that substrate deposition led to changes in meiofauna community composition and vertical distribution (Kaneko et al., 1997; Miljutin et al., 2011) which has also been observed in experiments on meiobenthic communities from shallower depths (Maurer et al., 1986; Mevenkamp et al., 2017; Schratzberger et al., 2000). However, it remains unclear which thickness of sediment deposition

evoked the meiofaunal response in the deep sea and if it is possible to reproduce the results under more controlled conditions on a short term.

Another possible risk of polymetallic nodule mining is the release of potentially toxic amounts of heavy metals during sediment resuspension and nodule abrasion with largely unknown effects on deep-sea biota (Hauton et al., 2017). Bioavailability and toxicity of metals inside marine sediments strongly depend on the structure and chemical properties of the substrate and these

complex processes are not yet fully understood (Eggleton and Thomas, 2004). While bulk sediment concentrations of heavy metals such as Cu, Ni and Cd are high in a polymetallic nodule area, the concentrations in the pore water, which constitute the potentially bioavailable fraction are significantly lower, ranging in the sub- or lower ppb level (Koschinsky, 2001; Paul et al., 2018). However, even if pore water concentrations of heavy metals are known, the effective uptake of those metals by infaunal organisms may still vary due to physiological adaptations to high metal burdens (Auguste et al., 2016). Therefore, direct

measurements of metals in animal tissues may be a more robust way to assess toxic effects induced by polymetallic nodule mining.

To evaluate the short-term effects of substrate burial on the structure of the meiobenthos community and metal uptake by nematodes, we deposited a 2 cm layer of crushed nodule substrate on enclosed, undisturbed abyssal sediments in the South-East Pacific at 4200 m water depth, using the remotely operated vehicle (ROV) *Kiel 6000*. Density and community structure

of the meiobenthos as well as the vertical structuring after eleven days of incubation was assessed in treatments with and without crushed nodule substrate deposition. Furthermore, nematode genus composition was investigated and X-ray spectrometric images were taken of nematodes to assess the usefulness of this technique for toxicity assessments in the deep-sea.



## 2    Material and Methods

### 2.1    Experiment set-up and sampling

The substrate burial experiment was performed *in situ* during RV *Sonne* cruise SO242-2 (28.08.2015 - 01.10.2015) at the southern reference site of the DISCOL experimental area in the Peru Basin, Southeast Pacific (7°7.51 S, 88°27.02 W, in

4196 m water depth; Thiel and Schriever, 1989). For this purpose the ROV *Kiel 6000* (GEOMAR, Germany) was used to insert six stainless steel rings (ø = 25 cm, height = 15 cm) into undisturbed sediment avoiding enclosure of nodules or megafauna. The rings were gently pushed 10 cm into the sediment until the collar around the rings touched the sediment surface (Figure 1 A). Subsequently, a substrate-distributing device (Figure S1) filled with 250 mL crushed nodule substrate was deployed on three of the steel rings (Burial treatment, Figure 1 B). The other three rings enclosed undisturbed sediments

which served as an experimental control (Control). Rotation of the T-handle activated the release of the substrate that was filled inside the tubes of the device. This resulted in a roughly homogenous distribution of crushed nodule substrate onto the sediment surface in the steel ring with a thickness of approximately 2 cm (Figure 1 C). After ~20 h, the sediment distributing devices were removed from the steel rings to allow complete settlement of all nodule particles and to ensure open water exposure during the remaining time of the experiment. After a total incubation time of eleven days, the sediment in each steel

ring was subsampled with two push cores (7.4 cm inner diameter, Figure 1 D).

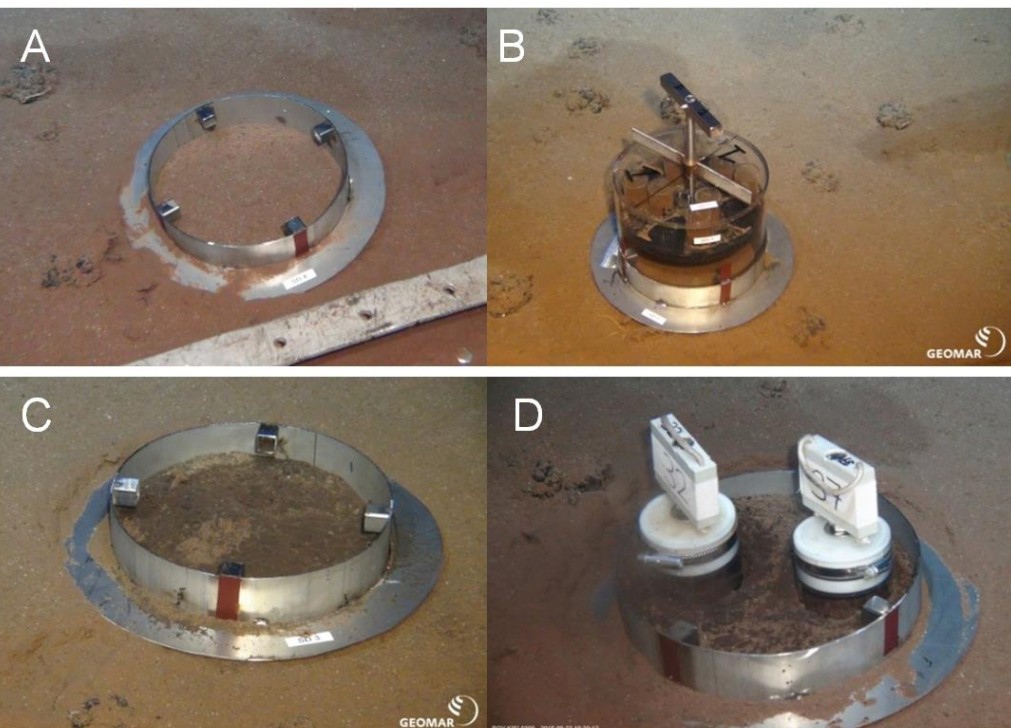

**Figure 1** Impressions of the deployment and sampling during the *in-situ* experiment. A) Stainless steel ring pressed into the abyssal sediment; B) Filled substrate-distributing device on top of the stainless steel ring before substrate release; C) Sediment surface after 11 days of incubation; D) Push core sampling at the end of the experiment. Copyright: ROV Kiel 6000 Team/GEOMAR, Germany.





On board, the overlying water in the push cores was carefully siphoned off and sieved (32 µm) to retain any meiobenthos. Subsequently, the sediment of each push core was sliced in several depth layers (added substrate layer (NOD), 0-1 cm, 1-2 cm and 2-5 cm sediment depth) in a climate-controlled room at *in situ* temperature (2.9 °C). The sediment of each slice from all

push cores was homogenized and a 5 mL subsample was taken for bulk sediment metal content analysis and stored at -20 °C. Of each set of push cores, one was used for meiobenthos community analysis and the sediment from each slice was fixed in 4% Borax buffered formaldehyde. The retained meiobenthos from the overlying water of that core was added to the sample of the uppermost sediment layer. The second push core served for the analysis of sediment characteristics (granulometry, total organic carbon content, total nitrogen content) and sediment from each slice was stored at -20 °C until further analysis.

Unfortunately, one core of the Control treatment was lost during slicing, leaving 2 replicates for environmental analyses.

## 2.2 Meiobenthos analysis

Meiobenthos sediment samples were washed on two stacked sieves of 32 µm (lower sieve) and 1000 µm (upper sieve). Meiofauna was extracted from the 32 µm fraction by density gradient centrifugation with the colloidal silica solution Ludox HS40 (specific gravity of 1.18)(Somerfield et al., 2005). After each of three centrifugation rounds (3000 rpm, 12 min), the

meiobenthos in the supernatant was retained on a 32 µm sieve. Subsequently, the sample was fixed in 4% buffered formaldehyde and stained with a few drops of Rose Bengal solution. Meiobenthos was identified to higher taxonomic level using a stereo microscope (50 x magnification).

From each sample, approximately 50 nematodes were picked, transferred stepwise to anhydrous glycerine following the formalin-ethanol-glycerol protocol of (De Grisse, 1969) and mounted on paraffin-ring glass slides for microscopic

identification. Nematodes were identified with a Leica DMLS compound microscope (10 x 100 x magnification) to genus level consulting mainly the original species descriptions and pictorial keys available on the Nemys website (www.nemys.ugent.be; (Bezerra et al., 2018). Furthermore, nematode genera were categorized in four feeding guilds based on their buccal cavity morphology as described by Wieser (1953). Feeding guilds included "selective deposit feeders" (Group 1A, small buccal cavity without teeth), "non-selective deposit feeders" (Group 1B, large buccal cavity without teeth), "epistrate feeders" (Group

2A, small buccal cavity with teeth) and "predators / scavengers" (Group 2B, lager buccal cavity with teeth). The mouthless genus *Parastomonema* was grouped separately ("mouthless").

## 2.3 Sediment characteristics and metal contents

Sediment grain size analysis was done by laser diffraction with a Malvern Mastersizer 2000 particle analyzer (Malvern Instruments, UK) and sediment fractions were classified according to Wentworth (1922). Total organic carbon (TOC) and total

nitrogen (TN) content in the sediments were analyzed with an Element Analyzer Flash 2000 (Thermo Fisher Scientific) after lyophilization, homogenization and acidification with 1 % HCl.





Sediment bulk concentrations of $Fe_2O_3$ (%), MnO (%), Cu, Ni and Co (ppm) were determined by inductively coupled plasma optical emission spectrometry (ICP-OES) following protocol nr. 14869-2:2002(E) of the International Organization for Standardization (2002). Pore water metal content was not measured in this study but has been assessed for the experimental area by Paul et al. (2018)

**2.4    Individual nematode copper content**

To determine copper contents in nematodes, respectively 6 and 11 similar-sized and shaped nematodes were taken from one sample of the added crushed nodule layer and the uppermost layer of a push core sample from the same experimental area taken during the incubation period and processed as described above. Nematodes were transferred to a drop of water and body length (L, µm, excluding filiform tail) and average width (W, µm, measured at three different positions in the middle body

region) were determined under a compound microscope connected to a Leica camera system. These measures were used to estimate nematode wet weight (WW) using an adjusted Andrassy (1956) formula to account for the specific gravity of marine nematodes (i.e. 1.13 g $cm^{-1}$): µg WW = L x $W^2$/1,500,000 (as described in Pape et al., 2013).

Nematodes were then mounted on 500 nm thin silicon nitride membranes (Silson Ltd, United Kingdom) by means of a small drop of MilliQ water and left to air-dry. Subsequently, element contents were assessed by means of micro X-ray fluorescence

(µXRF) using the Edax Eagle III (Edax Inc., USA). This instrument is equipped with a 50 W Rh X-ray tube fitted with polycapillary optics, which focus the X-ray beam in a 30 µm spot. A liquid nitrogen cooled Si(Li) detector is employed to capture the fluorescent X-rays. To examine the element content of the organisms, small mappings were performed with 30 µm step size; each measurement point of these mapping contains a full XRF spectrum with 10 s live time. These spectra are analysed using AXIL, an iterative least squares algorithm yielding the net intensities for each detectable element present in the

sample. The points belonging to the organism are extracted from the XRF element maps using k-means clustering. Next, the spectra from these data points are summed to obtain the total intensity generated by the nematode during the measurement. The intensities per nematode are normalized using nematode wet weights. Of the heavy metals, copper (Cu) was the only element visible clearly enough in the XRF spectra to yield reliable results. Due to the small diameter of the organisms (~ 30 µm) the absorption effects on Cu are negligible, so the normalized intensities of the different scans can be compared directly

with each other, in other words, a nematode with more Cu present in its body will yield a higher intensity (counts) per unit body mass (in µg).

**2.5    Data analysis**

Meiobenthos densities are expressed as the number of individuals per 10 $cm^2$ in the different depth layers and over the whole sampled depth (total densities). Due to the unequal thickness of the sampled depth layers, differences in community

composition were examined based on relative abundances of the different meiobenthic taxa in each depth layer.

K-dominance curves of nematode genera over the whole core were calculated based on untransformed density data (ind. 10 $cm^2$) and plotted in Primer 6 (Clarke and Gorley, 2006). Additionally, diversity indices (Shannon-Wiener, Pielou's evenness





and Simpson) of the whole core community were compared between treatments in univariate analyses. Differences in nematode genus composition between treatments and depth layers were analysed based on relative abundances, only.

Statistical differences between treatments and depth layers in multivariate datasets (sediment TOC and TN contents, meiobenthos community composition, nematode genus composition, nematode feeding types) were investigated with a cluster

analysis (cluster mode = group average) combined with a similarity profile test (SIMPROF) using Primer 6. For abiotic data, a resemblance matrix based on Euclidean distances was used while biotic data (meiobenthos and nematode genus composition, nematode feeding types) were analysed based on Bray-Curtis-similarities. Interpretation of the results was further based on a visualization with multidimensional scaling (MDS) plots and on the similarity percentages analysis (SIMPER) of significant cluster groups.

Differences of univariate measures (bulk sediment metal contents and total meiobenthos densities) between treatments were tested with a student's t-test in R (R Core Team, 2013) after ensuring normality (Shapiro-Wilk test) and homoscedasticy (Levene's test) of the data or, alternatively, with a Wilcox test as non-parametric test.

An $\alpha = 0.05$ significance level was chosen for all statistical analyses.

## 3    Results

### 3.1    Sediment characteristics and metal contents

The sediment in all push cores was characterized by a 10-20 cm thick brown layer of fluffy surface sediment with underlying more compact, whitish subsurface sediment and no differences in the coloration were apparent between the Control and the Burial Treatment at the end of the experiment (Figure S2).

The analysis of total organic carbon and nitrogen contents between treatments and depth layers revealed two significant clusters

branching at a distance of 0.4 ($\pi = 0.03$, p = 0.001). The first cluster was composed of all added substrate layers (NOD) and the second cluster contained all remaining samples. Differences were caused by lower TN and TOC contents in the crushed nodule layer (TN: $0.20 \pm 0.00$ %, TOC: $0.39 \pm 0.00$ %; mean $\pm$ standard error (SE)) compared to the Control (TN: $0.41 \pm 0.05$ %, TOC: $0.77 \pm 0.02$ %) and the underlying sediment layers (TN: $0.40 \pm 0.02$ %, TOC: $0.71 \pm 0.02$).

The Control sediment mainly consisted of silt ($75.6 \pm 0.2$ %; mean $\pm$ SE), clay ($12.8 \pm 0.2$ %) and very fine sand ($8.9 \pm 0.2$ %)

with a median grain size of $20.8 \pm 0.3$ µm, which was similar in the 0-5 cm of the Burial treatment (median grain size: $22.0 \pm 0.3$ µm). In contrast, the crushed nodule substrate contained much coarser grain fragments in the mm to cm range (Figure S3). Concentrations of Cu, Mn and Ni in the solid phase were more than three times higher in the crushed nodule substrate compared to the Control sediments (Figure 2).





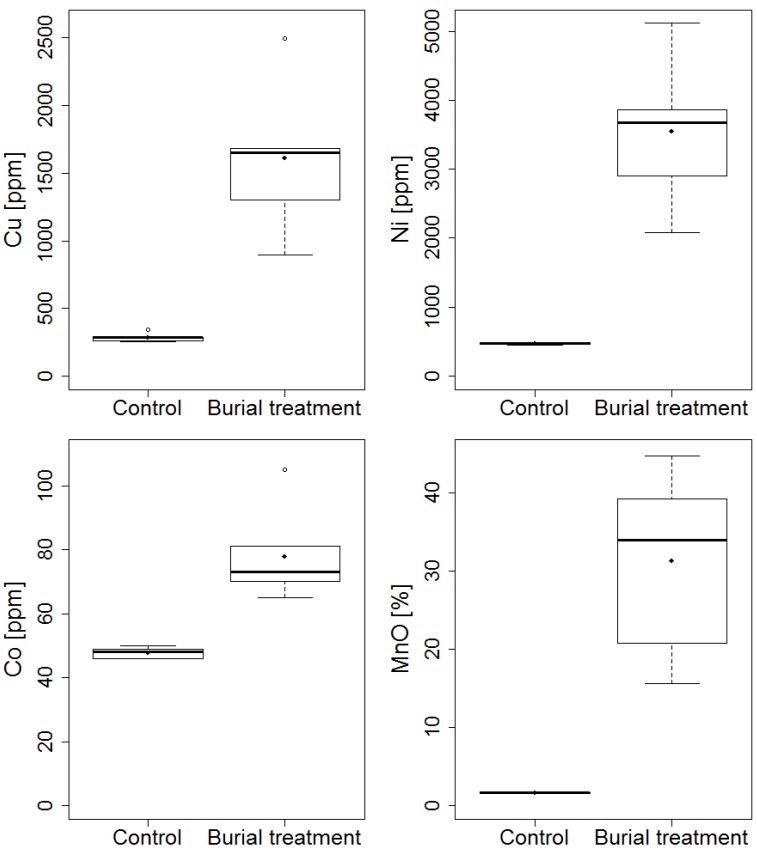

**Figure 2** Box-Whisker plot of solid-phase metal contents measured in the 0-1 cm layer of the Control and the added crushed nodule layer of the Burial treatment. Black line depicts the median whereas a filled dot indicates the mean of the measurements.

### 3.2    Meiobenthic community composition and vertical distribution

After 11 days of incubation, total meiobenthos densities ranged from $275 \pm 10$ ind. 10 cm$^{-2}$ (mean $\pm$ SE) in the Burial treatment to $303 \pm 24$ ind. 10 cm$^{-2}$ in the Control and did not differ between both treatments (Figure 3 A). Overall, nematodes dominated the meiobenthos community ($91.0 \pm 1.1$ %, mean $\pm$ SE) followed by harpacticoid copepods ($4.4 \pm 0.6$ %), nauplii ($3.2 \pm 0.7$ %) and polychaetes ($0.6 \pm 0.1$ %, Figure 3 B). All other taxa (Ostracoda, Tardigrada, Gastrotricha, Isopoda, Mollusca, Tantulocarida and Loricifera) contributed less than 0.5 % to the meiobenthos community.

In the Control, meiobenthos densities were similar across all depth layers with $40 \pm 3$ % of the meiobenthos occurring in the 0-1 cm layer, $28 \pm 5$ % in the 1-2 cm layer and $32 \pm 4$ % in the 2-5 cm layer (Figure 3 A). This vertical distribution was different in the Burial treatment where $46 \pm 1$ % of meiobenthos occurred in the added crushed nodule layer, $13 \pm 1$ % in the 0-1 cm layer, $10 \pm 1$ % in the 1-2 cm layer and $32 \pm 2$ % in the 2-5 cm layer (Figure 3 A). While at the end of the experiment $43 \pm 1$ % of nematodes over all depth layers were found in the added crushed nodule layer, this percentage was much higher

for polychaetes ($73 \pm 14$ %), copepods ($71 \pm 6$ %) and nauplii ($61 \pm 9$ %).



Whole core meiobenthos community composition was similar in samples of both treatments. However, when taking depth layers into account, two significant clusters were revealed branching at 92 % similarity ($\pi = 0.99$, p = 0.001). The first cluster (Cluster A) was composed of all crushed nodule layers (NOD), all 0-1 layers of the Control and one sample of the 1-2 layer of the Burial treatment (Figure 4) while the second cluster (Cluster B) was composed of all remaining samples. Similarities

5    between both clusters were caused by lower abundances of nematodes and higher abundances of copepods, nauplii and

polychaetes in Cluster A compared to Cluster B (SIMPER contributions: 48 %, 25 %, 15 % and 4 %, respectively; Table 1).

**Table 1** Results of the SIMPER analysis between the significantly different clusters identified in the dataset of relative meiobenthos abundances in different depth layers. Av.Abund = average abundance, Av.Diss = average dissimilarity, Diss/SD = average contribution divided by the standard deviation, Contrib% = Contribution to the dissimilarities, Cum% = Cumulative contribution to the dissimilarities.

|  | Cluster A | Cluster B |  |  |  |  |
| --- | --- | --- | --- | --- | --- | --- |
| Group | Av.Abund | Av.Abund | Av.Diss | Diss/SD | Contrib% | Cum.% |
| Nematoda | 87 | 95 | 4 | 3 | 48 | 48 |
| Harpacticoida | 7 | 2 | 2 | 3 | 25 | 73 |
| Nauplii | 4 | 2 | 1 | 1 | 15 | 88 |
| Polychaeta | 1 | 0 | 0 | 1 | 4 | 92 |

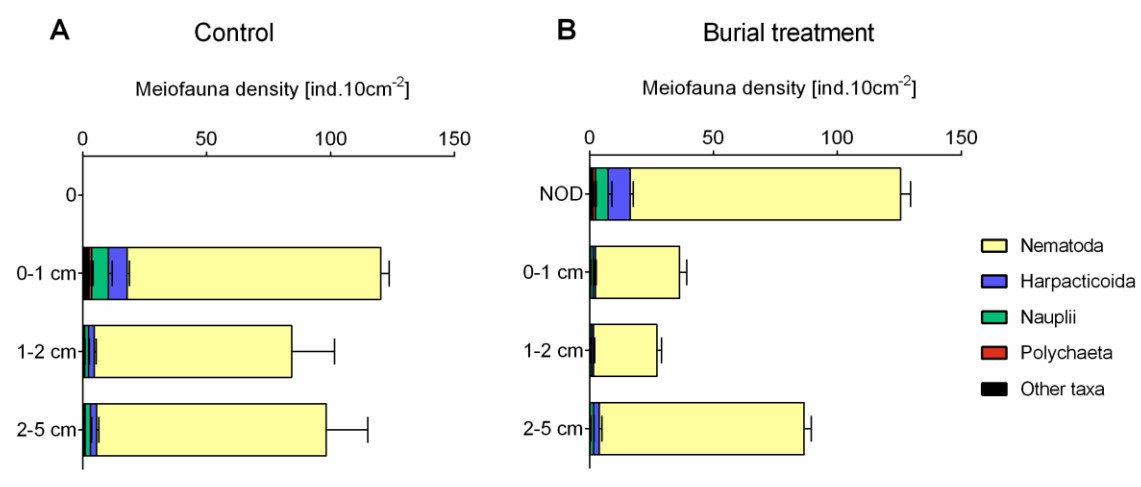

**Figure 3** Vertical profile of the average meiobenthos densities (ind. 10 cm⁻², + standard error) in the Control and Burial treatment with a ± 2 cm layer of crushed nodule substrate (NOD).



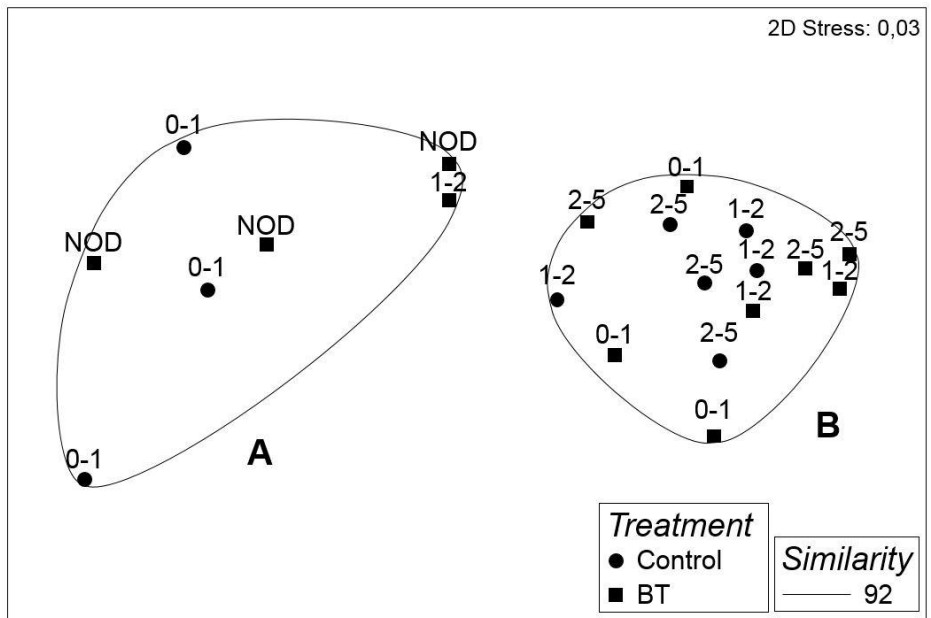

**Figure 4** MDS plot of the meiobenthos community in each sample of the Control and Burial treatment (BT) per sediment depth layer with overlying contours of significant (SIMPROF test) clusters at a 92 % similarity level indicated by the letters A and B. NOD = crushed nodule layer

## 3.3 Nematode genus community composition

The nematode genus community was very diverse and composed of 96 genera from 33 families combining all samples (Table S1). Of the total number of genera, 26 were only recorded once (singletons) and 18 were recorded twice (doubletons). The most dominant genera included *Acantholaimus* (14 ± 1 %), *Monhystrella* (11 ± 1 %), *Viscosia* (8 ± 3 %) and *Thalassomonhystera* (5 ± 1 %), of the other genera, each contributed less than 5 % to the overall nematode community. Evenness of nematode genera was higher in the Burial treatment (0.86 ± 0.01) compared to the Control (0.81 ± 0.01; $t_{2.73}$ = -3.373, $p$ = 0.0499, borderline significant, Figure 5). Diversity indices were not significantly different between the Burial treatment (Shannon: 3.23 ± 0.06, Simpson: 0.95 ± 0.01) and the Control (Shannon: 3.16 ± 0.08, Simpson: 0.93 ± 0.01, Figure 5).

The Cluster analysis of relative abundances of nematode genus composition revealed two significant clusters branching at 36 % similarity (π = 1.64, p = 0.002, Figure S4). However, due to the low similarity among all samples (likely resulting from the large number of rare genera) and the lack of clear groupings (e.g. samples of similar depth layers or treatments) within the clusters, we could not further interpret this result.





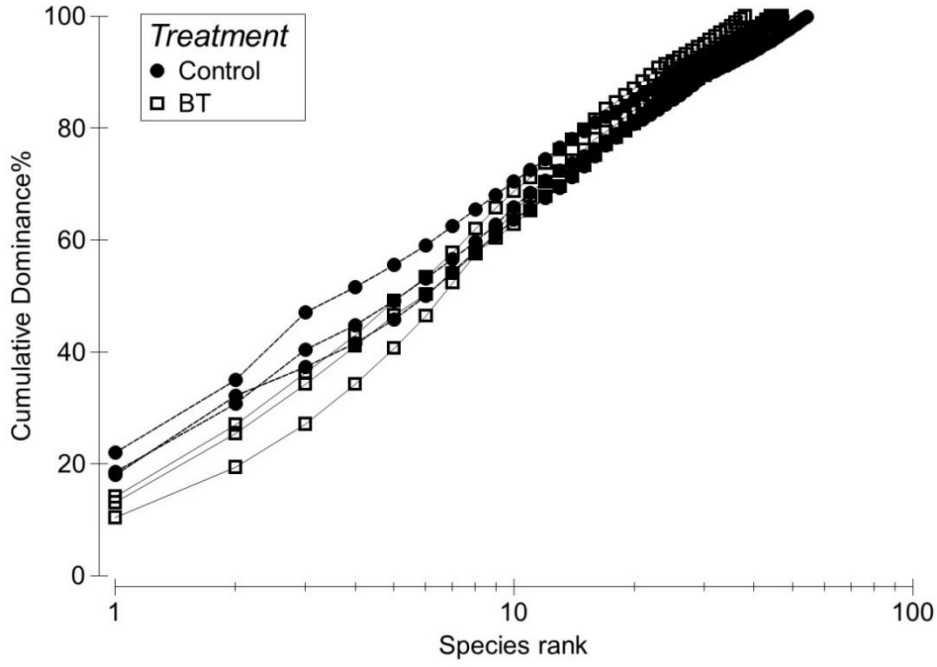

**Figure 5** K-dominance plot of whole core nematode genera in the Burial treatment (BT) and the Control.

When grouping genera into feeding types and, thereby, reducing variability between samples, the Cluster analysis of relative abundances of nematode feeding types resulted in 4 significantly different clusters. Cluster A included all crushed nodule

5  layers (NOD) and was different from cluster B which included most other depth layers from both treatments branching at a similarity of 75 % ($\pi$ = 1.45, p = 0.004, Figure 6). The two other clusters together included four depth layers of both treatments branching at 72 % and 46 % similarity (Cluster C and D, respectively, Figure 6).

SIMPER analysis indicated that the difference between the main clusters A and B was due to a reduction of non-selective deposit feeders by 9 %, an increase in epistrate feeders by 8 %, a reduction of selective deposit feeders by 4 % and an increase

10  in predators by 4% (SIMPER contributions: 37 %, 31 %, 16 % and 16 %, respectively, Figure 7).


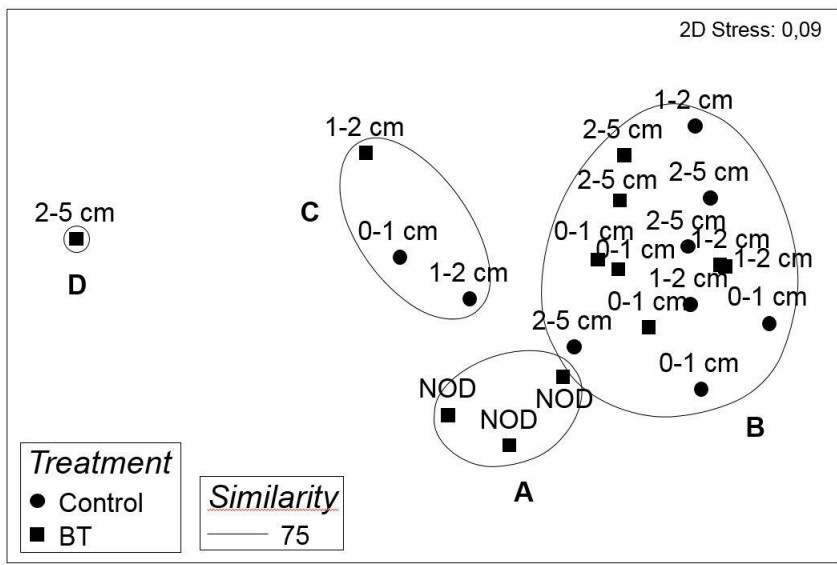

**Figure 6** MDS plot of the relative abundances of nematode feeding types in each sample of the Control and Burial treatment (BT) per sediment depth layer with overlying contours of significant (SIMPROF test) clusters at a 75 % similarity level indicated by the letters A – D. NOD = crushed nodule layer

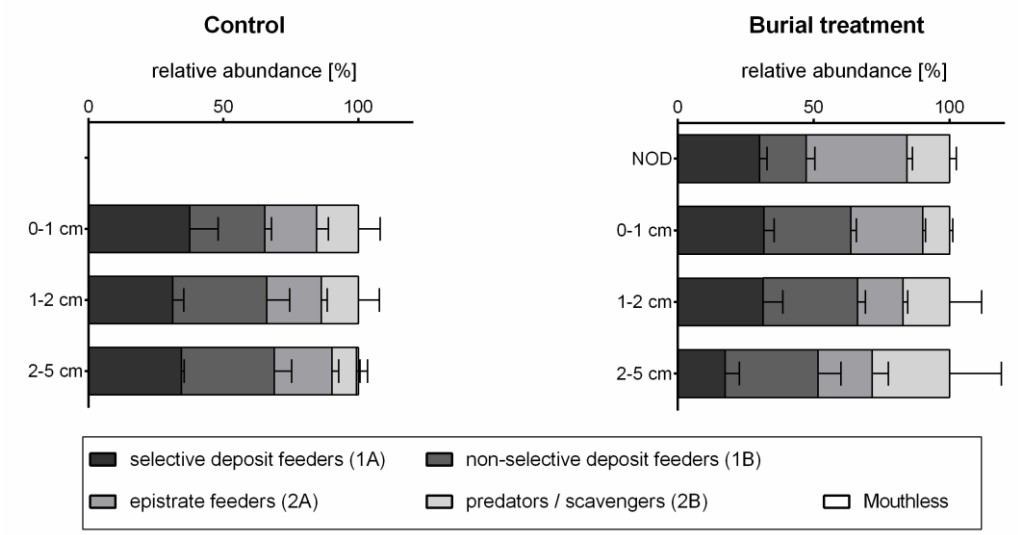

**Figure 7** Vertical profile of the relative abundance of nematode feeding types per sediment depth layer (percentage, + standard error) in the Control and Burial treatment with a ± 2 cm layer of crushed nodule substrate (NOD).

### 3.4 Copper burden in individual nematodes

Copper contents in nematode bodies could be successfully assessed using micro X-ray fluorescence (Figure 8 A). However,

10 copper burden in the measured nematodes did not differ between treatments (Figure 8 B).

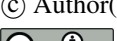



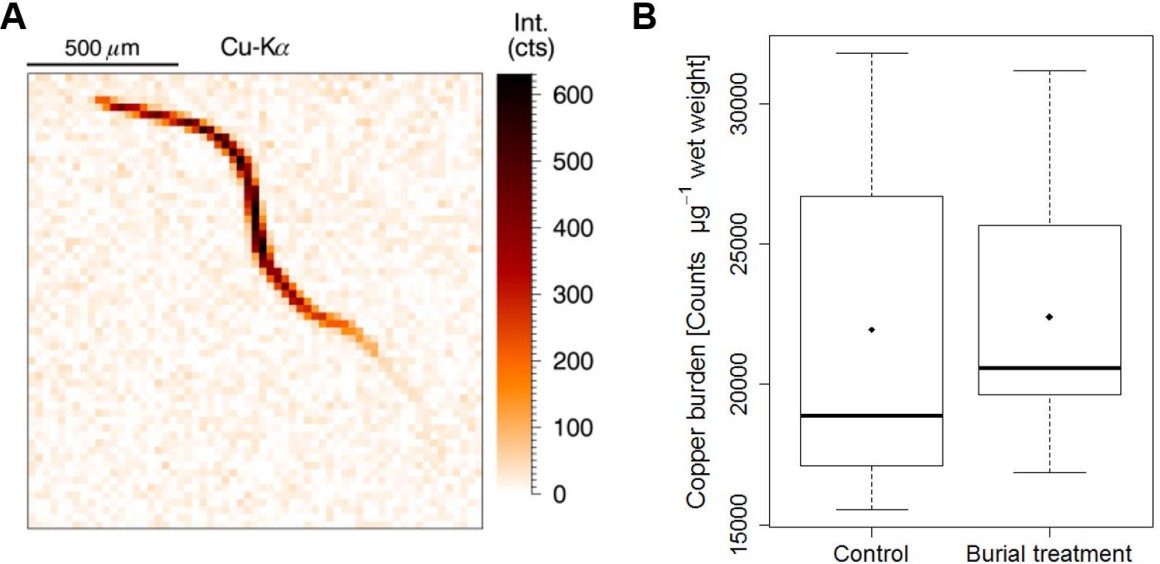

**Figure 8** A) Example image of the copper spectrum from a nematode X-ray mapping indicating copper intensity (counts), which is directly correlated to copper concentration. B) Box-whisker plot of the copper burden in nematodes from surface sediment layers of a background sample (n = 11) and from the crushed nodule substrate of the Burial treatment (n = 6). Black line depicts the median whereas dots indicate the mean of the measurements.

## 4    Discussion

### 4.1    Crushed nodule substrate burial induces changes in meiobenthos community structure

In a relatively short time span of eleven days, the meiobenthic community in our study responded to burial with crushed nodule particles by migration, adjusting their vertical position in the sediment. Almost half of the meiobenthos (46 ± 1 %), represented by all major taxa, had migrated into the added substrate layer at the end of the experiment. This migration was predominantly seen by fauna from the upper surface layers (0-2 cm) which showed strongly reduced densities compared to the same depth layers of the Control. We hypothesize that those organisms from the upper sediment layers are trying to re-establish their position in the sediment by moving upwards, while the mechanism that triggers this response remains unclear. Changes in oxygen penetration could be one such factor.

Migratory responses of meiobenthos have been widely observed and made use of in the past to deliberately extract these organisms from sediments, for example by changing temperature or oxygen availability to trigger meiobenthos movement (Rzeznik-Orignac et al., 2004; Uhlig et al., 1973). We were not able to monitor oxygen content over the time course of the experiment, but the burial with crushed nodule substrate could have led to reduced oxygen availability in the surface layer, causing the fauna to migrate upwards. In contrast to the surface fauna, meiobenthos densities in the deeper parts (2-5 cm) of



the sediment remained similar between the Control and the Burial treatments suggesting that the subsurface fauna is less sensitive to the changes in abiotic conditions causing the migratory response.

Changes in vertical nematode distributions have also been reported in a shallow-water study investigating the impacts of the disposal of experimental dredging material (Schratzberger et al., 2000) and a short-term laboratory experiment testing the

effect of instantaneous burial with inert tailings and dead, native sediment (Mevenkamp et al., 2017). These studies found that the amount and frequency of sediment burial are interactive factors showing that frequent but low amounts cause less severe changes than high amounts and instantaneous burial (Schratzberger et al., 2000); but also that substrate burial may cause nematode mortality of up to 50 % in the added substrate layer, which was measured by using a staining technique (Mevenkamp et al., 2017). Moreover, Mevenkamp et al. (2017) indicated that burial with 0.1 cm of tailings may already reduce the

functioning of bathyal, benthic fjord ecosystems in terms of fresh organic carbon remineralization. Especially, nematode uptake of added organic carbon was considerably reduced after burial with 0.5 cm of substrate. In contrast, Leduc and Pilditch (2013) report changes in vertical nematode distribution after experimental resuspension of the upper 5 cm of sediment originating from bathyal depths (345 m), but without marked effects on sediment characteristics or community oxygen consumption after 2 and 9 days following the disturbance. This different response may be attributed to the fact that the same

suspended sediment resettled in the experimental units, whereas Mevenkamp et al. (2017) investigated the deposition of additional substrate.

Interestingly, some meiobenthic groups including polychaetes, copepods and nauplii showed a stronger upward migration than nematodes in our experiment. Similarly, Kaneko et al. (1997) observed taxon specific responses to sediment burial in the Clarion-Clipperton Fracture Zone (CCFZ), a nodule region in the North-East Pacific. During the Japan Deep-Sea Impact (JET)

experiment, the authors created a sediment disturbance by means of a towed benthic disturber and sampled the area that was potentially impacted by re-sedimentation prior to, directly after and 1 year after the disturbance. In the collected upper 3 cm of the sediment, nematode densities were significantly reduced following the disturbance and remained low even after one year. In contrast, copepod densities remained similar directly after the disturbance and increased significantly after 1 year possibly due to their different mobility and life-history compared to nematodes. However, Kaneko et al. (1997) did not mention

the amount of resettled sediment and it is possible that the decrease in nematode densities was attributed to their limited upward migration directly after the disturbance (as was seen in our experiment) so that part of the nematodes were still residing in sediment layers >3 cm. While our experiment examined short-term and immediate responses to substrate burial, the JET experiment clearly indicated that sediment burial can also alter meiobenthic community structure on longer time scales.

The possibility that the migratory response may be accompanied by elevated mortality of abyssal meiofauna requires further

investigation, especially because re-sedimentation of fine clay is expected to occur over large areas in a deep-sea mining context (Oebius et al., 2001; Smith et al., 2008). Clay deposition could substantially change oxygen availability in comparison with the coarse nodule debris assessed here. In our experiment, we were not able to assess meiobenthic mortality resulting from the burial because decompression induced mortality during sample retrieval from abyssal depths would bias the results. Nevertheless, several authors have underlined the importance to assess meiobenthic mortality in short-term experimental



studies as it may pass unnoticed due to slow decomposition of organic matter in the deep sea (Barry et al., 2004; Fleeger et al., 2006, 2010). It should be noted that meiobenthic contribution to the benthic ecosystem in terms of relative abundance and biomass increases with water depth (Rex et al., 2006). Therefore, it is plausible that the induced changes in meiobenthos distribution -and, possibly, mortality- may entail even stronger effects on the overall functioning of abyssal soft sediments

with regard to food web interactions, organic matter remineralization and bioturbation.

## 4.2    Nematode community may face alterations in response to burial with crushed nodule particles

Generally, the abyssal seafloor is characterized by a low degree of disturbance and low organic matter input from the euphotic zone. Sedimentation rates in the Peru Basin are generally ranging between 0.4 and 2.0 cm ka$^{-1}$ (Haeckel et al., 2001). Therefore, in this environment we expect benthic assemblages that are adapted to very stable conditions. Interestingly, all dominant

nematode genera responded with upward migration and there was no evidence of specific selection mechanisms, e.g. opportunistic genera taking advantage of the new situation and being more successful in either inhabiting the new substrate or in remaining in the surface layers of the original sediment and, therefore, being more stress resistant. Opportunistic species generally occur under extreme, variable conditions and get outcompeted by less opportunistic species when disturbance is low (Grassle and Sanders, 1973). However, small scale disturbances and habitat heterogeneity in the deep sea may induce a more

dynamic environment to allow the persistence of colonizing species (Gallucci et al., 2005). This seems to be supported by the large number of Monhysteridae in our study, which are generally classified as good colonizers at least in shallow water environments (Bongers et al., 1991). However, deep-sea monhysterids are characterised by a high local intrageneric diversity not supporting an opportunistic behaviour (Vanreusel et al., 1997). The results of our experiment did not indicate that these nematodes were more successful to inhabit the added substrate.

Nematode communities in the abyssal deep-sea and nodule habitats in particular are characterized by a high diversity, potentially owing to the increased habitat complexity created by the nodules (Miljutin et al., 2011; Singh et al., 2014; Vanreusel et al., 2010). Similarly, the nematode community in our experiment displayed a high diversity with a large proportion of rare genera (45 % of the genera recorded only once or twice). This high proportion of rare taxa may increase the vulnerability of the nematode community in this area to disturbances since the risks of local extinction may be greater for those small

populations (McCann, 2000; Rosli et al., 2018) and recovery will depend on recolonization success and species connectivity. Information about these two factors is still very limited for nematodes from abyssal nodule regions, especially for the rare taxa. Our study indicated that the addition of crushed nodule substrate changed the relative abundance of feeding types in the new surface layer, which could, depending on the long-term effects (e.g. mortality, vertical restructuring, species interactions) affect the role of the nematode community in the functioning of the benthic environment.

A study of Miljutin et al. (2011) indicated that changes in nematode communities following strong sediment disturbance may persist for up to 26 years. They revisited a disturbed nodule site in the equatorial NE Pacific where sediment and nodules were removed by dredging 26 years ago and found that nematode density, diversity and community structure inside the disturbed track still differed from adjacent non-disturbed areas. This study clearly indicates that changes in nematode community




composition due to sediment disturbance, as seen in our experiment, may be long lasting and potentially irreversible with the risk of reducing local biodiversity through extinction of rare taxa.

### 4.3 Increased copper concentrations in the added substrate are not reflected in nematode body copper content

The very high concentrations of solid phase heavy metals in the crushed nodule substrate raise questions about bioavailability
and uptake of these metals in benthic organisms. Previous research in polluted, shallow waters has shown that nematodes play an important role in the transfer of heavy metals to and from the benthic food web in harbour communities (Fichet et al., 1999) and that uptake of different pollutants may vary (Howell, 1982). As such, Howell (1982) reported increased zinc uptake in nematodes exposed to pollution, while copper content was very variable and correlations with habitat pollution were less clear. In the presence of manganese (Mn) oxyhydroxides, other elements such as the transition metals (Cu, Ni, Zn) are adsorbed in
the oxic layer which is up to 5 and locally up to 20 cm deep in the Peru Basin (Stummeyer and Marchig, 2001). Therefore, most metals are bound to the solid phase of the sediment and are not bioavailable, which most likely explains our findings of similar copper burdens in the nematodes from the control and the Burial treatments. A recent study investigated the long-term and short-term effects of sediment disturbance on metal and trace element concentrations in the solid phase of the sediment and the pore water (Paul et al., 2018). The authors found that while some solid phase elements still deviated from pre-
disturbance levels even after 26 years, levels of trace metals in the pore water levels returned to pre-disturbance values in a short time frame of several weeks. Under conditions present in the sampled sediment, release of metals during a mining operation may not result in an increased metal toxicity because of the fast oxidation of Mn and absorption of metals (Paul et al., 2018). However, in the case that oxygen conditions change inside the sediment due to re-sedimentation or other processes the release of toxic compounds into the pore water cannot be excluded. The use of new techniques to analyse tissue metal
content as used in this study may allow precise detection of changes in heavy metal burden even in smaller benthic organisms due to mining related alterations of the abiotic environment.

### 4.4 Conclusion and recommendations

The brittle character of polymetallic nodules implies that the deposited material following mining will be a mixture of natural sediment and nodule particles. In this research, we revealed some important insights in the structuring of meiobenthic
communities following burial with freshly crushed nodule substrate. Despite the very different abiotic conditions in the crushed nodule substrate and the natural sediment in terms of grain size and carbon and nitrogen content, an upward migration of meiobenthic organisms was observed. Our results from *in-situ* experiments at >4000 m water depth confirm previous research in different habitats (Kaneko et al., 1997; Maurer et al., 1986; Mevenkamp et al., 2017; Schratzberger et al., 2000) showing that meiobenthic organisms generally show upward migration following burial with native and non-native substrate and
varying thickness of the deposited layer. We found that this behavioural response was stronger in polychaetes, copepods and their nauplii compared to nematodes, which could result in a shift in meiobenthos community composition.

Furthermore, the relative distribution of nematode feeding types was altered indicating that changes in the functional role of the nematode community on the short and long term cannot be excluded. Likely due to the high nematode genus diversity and evenness, changes in nematode genus composition were not detected between treatments.

The effect of vertical meiobenthos migration on other benthic size classes and over longer time scales requires further research, especially in a deep-sea mining context where sediment re-deposition is expected over large areas and long timescales (Murphy et al., 2016). Furthermore, we hypothesize that, although it is technically challenging, standardized methods for mortality assessments in deep-sea sediment samples are needed to advance our understanding of short-term environmental impacts on meiobenthos.

**Data availability:** The data used in this publication are deposited in the Pangaea database https://doi.pangaea.de/10.1594/PANGAEA.896027

**Author contribution statement**: AV, AB, KG and LM conceived the study. LM conducted the experimental work and collected the samples. Meiobenthos samples were analyzed by LM and KG, nematode copper content was measured and analyzed by BL and LV and sediment metal contents were provided by DV and JDG. LM performed the statistical data analysis; LM, BL, KG and AV interpreted the results and LM wrote the manuscript with the assistance of all authors.

**Acknowledgements:** The research leading to these results has received funding from the European Union Seventh Framework Programme (FP7/2007-2013) under the MIDAS project (grant agreement n° 603418) and was carried out with infrastructure funded by EMBRC Belgium (FWO project n° GOH3817N). The SO242-2 research cruise was funded by the German Ministry of Education and Science (BMBF, grant agreement n° 03F0707A-G) as a contribution to the European JPI Oceans Pilot Action "Ecological Aspects of Deep Sea Mining". LM received additional funding by the Flemish BOF research fund (BOF.DC1.2016.0006), and KG by the Flemish fund for Scientific Research (grant number: 1242114N). The authors want to thank the captain and crew of RV "Sonne", the team of ROV "Kiel 6000" as well as the scientist team on board. This publication reflects only the views of the authors; the EC is not liable for any use that may be made of the information contained herein.

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
