# Peer review of "Responses of an abyssal meiobenthic community to short-term burial with crushed nodule particles in the South-East Pacific"

_Biogeosciences, 2018_

## Referee Comment (RC1) · Sharma (Referee) · 22 Jan 2019

Journal: Biogeosciences (BG) Title: Responses of an abyssal meiobenthic community to short-term burial with crushed nodule particles in the South-East Pacific Author(s): Lisa Mevenkamp, Katja Guilini, Antje Boetius, Johan De Grave, Brecht Laforce, Dimitri Vandenberghe, Laszlo Vincze, and Ann Vanreusel MS No.: bg-2018-489 MS Type: Research article Special Issue: Assessing environmental impacts of deep-sea mining – revisiting decade-old benthic disturbances in Pacific nodule areas

General comments:

[Figure]

The study by Lisa Mavenkamp et al. on response of abyssal meiobenthic community to burial with nodule particles adds a new dimension to the understanding of possible responses of deep-sea faunal groups on mining of seabed minerals. The methodology adopted by the authors is unique with interesting results and deserves to be published. However, the authors need to take into consideration the following:

1. Describe in 'Introduction' as to what is the likely source of the crushed nodules during a mining operation and what is the expected concentration and size of the nodule particles that are will be introduced on the seafloor based on which this experiment is planned. Also mention what is the size and concentration of the crushed nodules used in the experiment.

2. The study makes certain comparisons with the results of previous benthic impact experiments (BIEs) that were not at all similar either in spatial terms or time scales or volume and nature of resedimentation, This study is based on effects of concentrated crushed nodules on meiobenthos over eleven days in a restricted area, whereas the BIEs were studies of impacts of distribution of resuspended sediments (and not crushed nodules) over large areas and longer periods of time (1 year or more). So, it is not correct to compare the results of these two.

3. It is interesting to see that many of the results of this experiment have shown positive response of meiofauna as well as other groups (upward migration into the resedimented layer, no additional accumulation of copper, and no extreme changes in community structure) that should be highlighted. When the understanding of likely impacts of deep-sea mining is limited and mostly negative impacts are being projected by the environmental groups based on little or no data, it is important to bring out positive impacts as well so as to have a balanced approach towards sustainable mining. Also researchers need to appreciate that it is not necessary that all responses to any man-made activity will always be negative, but could be positive as well as shown in this study and this is an important contribution from the marine biologists to this subject.

Specific comments:

1. Page 1, line 23 - Abstract - change '....in covered and undisturbed sediments.' to '....in covered and uncovered sediments.' (because there is no other disturbance on the seafloor but covering of sediments by crushed nodules).

2. Page 2, line 6 – Introduction – change 'extraction in' to 'exploitation from' – as 'extraction' means 'removal of metals from ore', whereas 'exploitation' means 'removal of ore from its original source".

3. Page 2, line 14 – change 'such as' to 'due to'- as these are causes not impacts.

4. Page 4, line 8 – Correct 'Fig S1" to 'Fig 1',

5. Page 4, line 8 – change 'substrate distribution device' to 'crushed nodule distribution device' – see below for explanation.

6. Page 4, line 8 – According to Cambridge dictionary, the word 'substrate' means something lying below or base or bed and cannot be used for crushed nodules being deposited artificially from top. So change 'crushed nodule substrate' to ' crushed nodule particles' and 'substrate' to 'nodule particles' in the entire manuscript..

7. Page 4, line 10-11 – Add mean size of crushed nodules '...substrate of ### micron / mm size that was filled inside the tubes of the device.'

8. Fig. 1 caption – change 'Impressions' to 'Images' or 'Photographs'

9. Page 7, line 14 – Please mention the 'values for sediment characteristics, metal values, and meiofauna composition' before the experiment and compare the values after the experiment to evaluate the impact of burial of seafloor sediment by crushed nodule particles.

10. Page 7, line 25 - change 'burial treatment' to ' burial treatment sediment sample'.

11. Page 9, line 3 – add units 'cm' after '0-1' and '1-2'.

12. Page 10, line 7 – change 'Table S1' to 'Table 1'

13. Page 10, line 10 – change 'control' to 'control samples' and 'burial treatment' to 'burial treatment samples'.

14. Page 10, line 15 – change 'Figure S4' to 'Figure 4'.

15. Page 13, line 14-15 – After 'Changes in oxygen could be one of the factor' it would help to give either layer-wise oxygen values before and after the experiment or at least give general values for oxygen content in sediment layers from other publications (eg. Rzeznik-Orignac et al, 2004) to support the hypothesis.

16. Page 15, line 9-10 – 'Interestingly, all dominant nematode genera responded with upward migration…..' is a positive response further supported by section heading 4.3 'Increased copper concentrations in added substrate are not reflected in nematode body copper content' - these need to be highlighted as mentioned in #3 of general comments.

17. Page 15, line 18-19 – 'The results of our experiment did not indicate that these nematodes were more successful to inhabit added substrate' contradicts the above statements, unless it refers specifically to monhysterids. So please specify or remove this sentence.

18. Page 15, line 30-34 – Effects of resedimentation of sediment over large open area and that of crushed nodules over small enclosed area cannot be compared as the material deposited and the process and concentrations are entirely different. The discussion needs to be modified.

19. As the experiment of depositing crushed nodules has shown positive response of nematodes by upward migration and maintaining similar community structure, the sentence on page 16 (line 1-2) '….Changes in nematode composition…. may be long lasting and positively irreversible…..' need to be revised.

20. Page 16 – Conclusions – needs to be revised according to the above discussions.

Recommendation: It is recommended that major revision is required before accepting the paper.

---

## Referee Comment (RC2) · Anonymous Referee #2 · 24 Jan 2019

The manuscript describes results from an experiment to assess the combined effect of burial and manganese nodule particles on abyssal meiofaunal communities. I though the manuscript was very interesting, and written by a rising star in deep-sea ecology. The paper and data will be very useful to academics as well as policy organisations dealing with the effects of sediment and nodule particle deposition from deep-sea mining for polymetallic nodules. My main concern about the manuscript is that the substrate addition didn't appear to have a huge impact on benthic community structure in the experiments. While these are the results that have been collected and need to be reported, my feeling is that a lot of the fauna in the substrate addition treatment were actually dead but hadn't decomposed at the end of the experiment. Then, when

the fauna were preserved in formalin after 11 days everything that was alive and dead at the end of the experiment was preserved such that no change in community composition could be detected. I understand that this is difficult to assess using staining methods (as stated in the discussion by the author), but it would have been possible to assess the condition of some of the meiofauna at the end of the experiment (e.g., by looking at the appearance of the striated-muscles of the harpacticoids from the burial treatment, and comparing with the control samples). Similar approaches have been undertaken in the past (see Thistle et al. 2005 , Mar. Ecol. Prog. Ser. 289: 1-4) to estimate the proportion of meiofaunal harpacticoids killed in situ by $CO_2$ perturbations. I would suggest that the lack of information about meiofaunal death is clearly flagged as a possible reason why differences in benthic community composition could not be detected. Although the authors went some way to discuss meiofaunal death in their discussion, this point really needs to be stressed. This is because, at present, mining contractors may use this paper to state that manganese nodule particle/ sediment deposition does not alter benthic community composition, and I am not convinced this will be the case.

I recommend that the article be published eventually following some moderate revisions.

Minor points to consider: Abstract 1) Line 11: change to "may rive the extraction of deep-sea mineral. . ." 2) Line 13: Change to "Experimental studies are scarce and simulated effect studies are small scale relative to the effects that will be seen during deep-sea mining. . ." 3) Line 16: Insert "in 2015" after conducted. 4) Line 22: Remove "original" Introduction: 1) Page 2, Line 10: It would be good to provide the range of typical manganese nodule growth rates here, because <250mm myr-1 can mean 0.00000000001mm myr-1 to 250mm myr-1. 2) Page 2, Line 15: What about organic matter dilution as well 3) Page 2, Line 18: change "of" to "from" 4) Page 2, Line 22: It would be good to give the reader some idea about the natural sedimentation rates in the abyss, and some indication of the levels of sedimentation that will occur during deepsea mining. 5) Page 3, line 2: Change to "which causes at worse, meiofaunal death, but at least removal..." 6) Page 3, line 24-26: I am confused as to why the amount of metal in the animal tissues is a robust way to assess toxic effects. You could have an animal with a high level of metals in its tissues, but the animal is highly resilient to metal toxicity. Therefore, the amount of metal in its tissue does not really always show the degree of toxicity from that particular metal. Methods: Overall methods. Did you assess the volume of the sediment taken up by solid nodule particles in your 10cm2 sample from the controls and burial treatments. If some sediments have more solid nodule particles, then there is less sediment to inhabit and this may have an effect on the densities that you found. 1) Page 4, line 8: You need to mention how you sampled the nodule and crushed the nodule to make the substrate. This information is missing. 2) Page 5, line 8: Change to "The second push core was used to..." 3) Page 5, line 9: Did you try and get an idea of the organic matter quality of the sediment and the added substrate? Given that a lot of meiofauna directly consume labile microbial organic matter (see Bernhard & Bowser. 1992. Mar. Ecol. Prog. Ser. 83: 263-272, Ingels et al. 2010. Mar. Ecol. Prog. Ser. 406: 121–133.), the quality of the substrate, as well as the effects from burial and the content of the manganese substrate may have all had an influence on the meiofaunal response. If you do not have actual Chl-a, or lipid concentrations, you can at least get an idea from the C: N ratio. 4) Page 6, line 18: Please define "live time". It sounds cool, but I have no clue what this is. 5) Page 7, line 1: What Simpson metric are you referring to? The term 'Simpson's' can actually refer to any one of 3 closely related indices (Simpson's Index, Simpson's Index of Diversity or Simpson's Reciprocal Index). 6) Page 7, line 1: What univariate analyses were used? Results: 1) Page 10, line 13: I think that the biodiversity metrics being the same in both the burial and control treatments may be due to you not being able to differentiate between live and dead fauna. This could have been assessed in the harpacticoids by looking at the condition of the fauna, since dead fauna would appear more degraded even if they've been at abyssal temperatures for a few days. As I stated before, it is important that the manuscript is carefully worded to reflect this as this result could be used as evidence for

no impact from re-sedimentation of sediment and nodule particles during mining, and I doubt this will be the case given the low background sedimentation rates in the abyss. 2) Regarding my first point in the methods section above, it would probably have been a good idea to standardise your meiofauna abundances to per unit volume of sediment rather than area. If the nodule substrate layer was full of cm-sized particles then the amount of living space available to the nematodes would be significantly less than in the control samples. Standardising the abundances to unit volume (if you have the data) may show much larger differences, and you may detect differences in community structure, or abundance (at least) between treatments. Discussion: 1) Page 13, line 19: Given the coarse nature of the nodule particles, wouldn't $O_2$ penetrate through the manganese substrate layer relatively easily. I understand there is burial, but diffusion will be dependent on the porosity, which should be greater in the substrate layer. 2) More overall impression of the discussion is that the authors need to acknowledge the weaknesses of the study (e.g., being unable to document meiofauna death) to a much better degree. This is done somewhat, but it really needs to be emphasized that a lot of the responses seen (or lack of them, e.g., in the biodiversity data) may be caused by the inability to distinguish living from dead fauna in the different treatments.

---

## Referee Comment (RC3) · Anonymous Referee #3 · 28 Jan 2019

General comments: This study explores a really challenging question, that of how deep sea meiofauna respond to mining operations. It is an increasingly vital question as we learn more about the diversity and importance of deep sea meiofauna and as deep sea mining operations expand. I applaud the authors efforts to tackle this problem and I think this study should be published but with some clarification and moderate revisions.

My biggest issues with the article center around their methodology and interpretation of depositing nodule sediment onto existing sediment. First, there is no indication that I can see of where this nodule sediment came from? How far from the "regular" sediment

on which it was deposited was it collected? Are these nearby habitats or hundreds of miles apart? Also, why did the authors choose to deposit nodule sediment alone when in their description of nodule mining practices it seems that there is removal of nodule sediment, disturbance of underlying or neighboring sediment, and deposition of nodule particles mixed with suspended sediment and redeposited. It seems the mining operations are after the nodule sediment in particular, so why would they ever redeposit it onto non-nodule sediment? Unless by accident? Please clarify where the nodule sediment came from and why its direct deposition onto non-nodule sediment was chosen as the primary methodology as this doesn't seem to mimic any aspect of the mining operations under question.

Also, the authors mention (with citations) in the discussion that meiofauna does inhabit the nodule sediment, yet there seems to be no taking this into account when interpreting the behavior of the meiofauna upon burial. Was the nodule sediment sterilized? Was it presumed that the meiofauna washed out upon transport? It seems like the primary interpretation of the presence of meiofauna in the nodule sediment at the end of the study is that it was colonized from the buried sediment below due to upward movement, but couldn't there have been a meiofaunal community in the nodule sediment upon deposition? If you didn't remove the meiofauna or examine it beforehand, how do you know that meiofauna found in it afterward came from the buried sediment?

Specific comments:

Page 2, line 1: "70s" should be "1970's"

Page 2, line 17: Here is where you describe mining operations and what happens to nodule sediment and "regular" sediment. You even mention how a large scale mining operation "is expected to directly impact the nodule associated fauna" so then it seems confusing that you then proceed to assume there is no fauna there until you place it on other sediment in your study.

Page 4, line 8: Please specify here where the crushed nodule substrate came from
and how/if it was treated.

Page 7, line 1: Citations would be helpful for all of these diversity indices and to specify which Simpson index.

Page 13, line 10: Please clarify here why you think that all the meiofauna in the nodule sediment came from the lower layer ("adjusting their vertical position in the sediment").

Page 14, line 25: Here you mention a study that showed a decrease in nematode densities "attributed to limited upward migration directly after the disturb (as was seen in our experiment)..." but previously you had indicated that there was considerable vertical migration from the lower sediment. Please clarify this.

Page 15, line 27: Here you indicate that your study found that the addition of crushed nodule substrate "changed the relative abundance of feeding types in the new surface layer..." yet you don't seem to have examined the nematodes in the surface layer (nodules) before depositing it, so how can you know this?

---

## Author Comment (AC1) · 7 Mar 2019

We are very thankful for the thorough and constructive comments and remarks on our manuscript made by Dr. Sharma. The issues raised by the reviewer were taken into consideration and in the following paragraphs, we present our reply to each of them:

General remarks:

Reviewers comment: 1. Describe in 'Introduction' as to what is the likely source of the crushed nodules during a mining operation and what is the expected concentration and size of the nodule particles that are will be introduced on the seafloor based on

which this experiment is planned. Also mention what is the size and concentration of the crushed nodules used in the experiment.

Authors reply: The most likely source of the spreading of nodule particles will be the collecting device. Nodule particles may be abraded and brought into suspension by the water jet used during the collection or, depending on the design of the collector, during separation of nodules and sediment inside the collector. This will result in the distribution of nodule particles, which are smaller than the mesh used for the separation. These particles would then spread as part of the sediment plume and would settle depending on their sedimentation rate. Particle sizes used during this experiment are shown in Supplementary figure S2 (former S3) and the material and method section has been expanded (see specific comment #7)

Changes to the manuscript: Page 2, Line 13: We expanded the sentence to "Therefore, breakage and abrasion of nodule particles is likely to occur during a mining operation with heavy gear, for example during separation of nodules and sediment as part of the collection process or by the force of the water jet used for the collection of nodules."

Reviewers comment: 2. The study makes certain comparisons with the results of previous benthic impact experiments (BIEs) that were not at all similar either in spatial terms or time scales or volume and nature of re-sedimentation. This study is based on effects of concentrated crushed nodules on meiobenthos over eleven days in a restricted area, whereas the BIEs were studies of impacts of distribution of resuspended sediments (and not crushed nodules) over large areas and longer periods of time (1 year or more). So, it is not correct to compare the results of these two.

Authors reply: We agree with the reviewer in the sense that a direct comparison I indeed not very relevant to compare our results with the JET experiment and we therefore would opt to remove that paragraph from the discussion. Nevertheless, migratory responses by meiofauna have been observed in several studies regardless of the nature of the substrate used. Therefore, we would argue that, to a certain extent, it is

valid to use studies using different material to interpret our findings, while considering also the differences in approach.

Changes to the manuscript: The comparison with the JET experiment on Page 14, Line 17-28 was removed.

Reviewers comment: 3. It is interesting to see that many of the results of this experiment have shown positive response of meiofauna as well as other groups (upward migration into the re-sedimented layer, no additional accumulation of copper, and no extreme changes in community structure) that should be highlighted. When the understanding of likely impacts of deep-sea mining is limited and mostly negative impacts are being projected by the environmental groups based on little or no data, it is important to bring out positive impacts as well so as to have a balanced approach towards sustainable mining. Also researchers need to appreciate that it is not necessary that all responses to any manmade activity will always be negative, but could be positive as well as shown in this study and this is an important contribution from the marine biologists to this subject.

Authors reply: We understand the concern raised by the reviewer, however, we do not believe that the changes in vertical distribution should be described as positive. Indeed some responses, such as copper accumulation were absent/neutral but the upward migration of the meiofauna from upper layers and changes in feeding type composition should be interpreted with care as long-term effects are unknown and may not necessarily be positive. It is not our intention to highlight negative effects of deep-sea mining but to interpret our findings based on the available information. We rephrased some sentences in the manuscript that may be interpreted in a stronger way than intended.

Changes to the manuscript: Page 1 Line 28-29 "The results indicate that short-term substrate burial requires special attention with regard to ecological consequences of mineral extraction in the deep-sea." Was changed to "Our results indicate that shortterm impacts from burial with crushed nodule particles on meiobenthic communities are limited but that long-term studies are needed, especially with regard to vertical structure, community composition and mortality."

Page 11, Line 30-31 the sentence "We found that this behavioural response was stronger in polychaetes, copepods and their nauplii compared to nematodes, which could result in a shift in meiobenthos community composition." was removed.

Specific comments

Reviewers comment: 1. Page 1, line 23 - Abstract - change '....in covered and undisturbed sediments.' To '....in covered and uncovered sediments.' (because there is no other disturbance on the seafloor but covering of sediments by crushed nodules).

Authors reply: adjusted as suggested

Reviewers comment: 2. Page 2, line 6 – Introduction – change 'extraction in' to 'exploitation from' – as 'extraction' means 'removal of metals from ore', whereas 'exploitation' means 'removal of ore from its original source".

Authors reply: adjusted as suggested

Reviewers comment: 3. Page 2, line 14 – change 'such as' to 'due to'- as these are causes not impacts.

Authors reply: adjusted as suggested

Reviewers comment: 4. Page 4, line 8 – Correct 'Fig S1" to 'Fig 1',

Authors reply: throughout the manuscript, the label of supplementary figures was adjusted to "Supplementary Fig. S..." to avoid confusion with the figures inside the manuscript.

Reviewers comment: 5. Page 4, line 8 – change 'substrate distribution device' to 'crushed nodule distribution device' – see below for explanation.

Authors reply: see next reply

Reviewers comment: 6. Page 4, line 8 – According to Cambridge dictionary, the word 'substrate' means something lying below or base or bed and cannot be used for crushed nodules being deposited artificially from top. So change 'crushed nodule substrate' to ' crushed nodule particles' and 'substrate' to 'nodule particles' in the entire manuscript..

Authors reply: We would like to keep the phrasing "substrate" ins the manuscript. The added material is intended to be used as a new substrate by the fauna after deposition, similar to sediment, and relates to the Cambridge dictionary definition "a substance or surface that an organism grows and lives on and is supported by". This wording is particularly useful when referring to different material without the need to specify each material separately (e.g. nodule particles, inert tailings, sediment).

Reviewers comment: 7. Page 4, line 10-11 – Add mean size of crushed nodules '...substrate of ### micron / mm size that was filled inside the tubes of the device.'

Authors reply: Mean sizes of the nodule substrate were not measured and would not be very informative due to the large range of the particle sizes. The Material and Method section was expanded with information about acquisition of the nodule particles including size range: Page 4 Line 15 "To obtain the crushed nodule particles, several nodules from the experimental site were collected 2 days prior to the experiment. Upon retrieval, epifauna, if present, was removed from the nodules and nodules were thoroughly washed with fresh water to remove all sediment and fauna. Subsequently, nodules were put inside plastic bags and manually crushed with a hammer. The resulting nodule particles varied in size between 3 $\mu$m and 1 cm (Supplementary Figure S2)."

Reviewers comment: 8. Fig. 1 caption – change 'Impressions' to 'Images' or 'Photographs'

Authors reply: "Impressions" changed to "Images"

Reviewers comment: 9. Page 7, line 14 – Please mention the 'values for sediment characteristics, metal values, and meiofauna composition' before the experiment and compare the values after the experiment to evaluate the impact of burial of seafloor sediment by crushed nodule particles.

Authors reply: We did not conduct a sampling of the sediment before the experiment. Results are based on a Control-Treatment comparison. However, also in the Control, stainless steel rings were used to achieve the same conditions (e.g. limiting lateral movement, water flow) in both treatments.

Reviewers comment: 10. Page 7, line 25 - change 'burial treatment' to ' burial treatment sediment sample'.

Authors reply: adjusted as suggested

Reviewers comment: 11. Page 9, line 3 – add units 'cm' after '0-1' and '1-2'.

Authors reply: units were added

Reviewers comment: 12. Page 10, line 7 – change 'Table S1' to 'Table 1'

Authors reply: "Table S1" was changed to "Supplementary Table S1"

Reviewers comment: 13. Page 10, line 10 – change 'control' to 'control samples' and 'burial treatment' to 'burial treatment samples'.

Authors reply: In our view, this change does not significantly add to the understanding of the sentence and this distinction would need to be applied also to all other sentences.

Reviewers comment: 14. Page 10, line 15 – change 'Figure S4' to 'Figure 4'.

Authors reply: "Figure S4" was changed to "Supplemenary Figure S4" Reviewers comment: 15. Page 13, line 14-15 – After 'Changes in oxygen could be one of the factor' it would help to give either layer-wise oxygen values before and after the experiment or at

least give general values for oxygen content in sediment layers from other publications (eg. Rzeznik-Orignac et al, 2004) to support the hypothesis.

Authors reply: Unfortunately, as mentioned on Page 13 Line 18 we were not able to measure oxygen concentrations in our sample. We hypothesized oxygen availability as explanation for the observed behavior as it has been proposed to play a role in meiofauna extraction techniques. Furthermore, oxygen penetration depth was reduced in the experiments of Mevenkamp et al 2017 leading the authors to hypothesize this oxygen reduction as an explanation for the upward migration of nematodes and increased mortality.

Changes to the manuscript: "by using natural gradients of oxygen availability" was inserted at Page 13, Line 17

Page 13 Line 17 "Moreover, in a short-term laboratory experiment, Mevenkamp et al. (2017) observed significantly reduced oxygen concentration in the underlying soft sediment after the addition of 0.5 and 3 cm sediment and an upward migration and increased mortality of nematodes." was added

Reviewers comment: 16. Page 15, line 9-10 – 'Interestingly, all dominant nematode genera responded with upward migration.....' is a positive response further supported by section heading 4.3 'Increased copper concentrations in added substrate are not reflected in nematode body copper content' - these need to be highlighted as mentioned in #3 of general comments.

Authors reply: See reply on general comment #3.

Reviewers comment: 17. Page 15, line 18-19 – 'The results of our experiment did not indicate that these nematodes were more successful to inhabit added substrate' contradicts the above statements, unless it refers specifically to monhysterids. So please specify or remove this sentence.

Authors reply: Indeed, this sentence refers to the monhysterids. "these nematodes"

replace with "monhysterids"

Reviewers comment: 18. Page 15, line 30-34 – Effects of resedimentation of sediment over large open area and that of crushed nodules over small enclosed area cannot be compared as the material deposited and the process and concentrations are entirely different. The discussion needs to be modified.

Authors reply: We changed the last sentence to acknowledge the difference in the disturbance between both studies and to not mislead the reader. Nevertheless, we believe that the persisting change in nematode communities observed by the cited study should be mentioned here to underline the caution needed in the interpretation of disturbances in the deep-sea as even small changes may be long-lasting and a no-impact-conclusion should not be drawn too fast.

Changes to the manuscript: Page 15 Line 33 sentence adjusted to "Although the disturbance studied by Miljutin et al. (2011) strongly differs from our experiment, it indicates that changes in nematode community composition in polymetallic nodules areas may be long lasting and are potentially irreversible and, therefore, underlining the importance of long-term experiments."

Reviewers comment: 19. As the experiment of depositing crushed nodules has shown positive response of nematodes by upward migration and maintaining similar community structure, the sentence on page 16 (line 1-2) '....Changes in nematode composition.... may be long lasting and positively irreversible.....' need to be revised.

Authors reply: Community structure on a higher taxonomic level was indeed similar in both treatments, but at lower taxonomic level, changes in nematode feeding types were observed. Therefore, we do believe that this sentence is still valid in order to alert the reader on the potential long-term risks. Furthermore, samples exhibited a very high diversity and high evenness with many rare (<5%) taxa, also evidenced by the low similarity among replicates, which may increase the risk of losing rare taxa after strong sediment disturbances. Nevertheless, in response to the comment #18, the sentence

was rephrased (see above).

Reviewers comment: 20. Page 16 – Conclusions – needs to be revised according to the above discussion

Authors reply: This comment relates to our reply to general comment #3. We have done small adjustments to the text; however, also taking into account the remarks of reviewer 2, we would like to keep the overall conclusion as it is.

―――――――――――――――――――――――

---

## Author Comment (AC2) · 7 Mar 2019

Reviewers comment: The manuscript describes results from an experiment to assess the combined effect of burial and manganese nodule particles on abyssal meiofaunal communities. I though the manuscript was very interesting, and written by a rising star in deep-sea ecology. The paper and data will be very useful to academics as well as policy organisations dealing with the effects of sediment and nodule particle deposition from deep-sea mining for polymetallic nodules. My main concern about the manuscript is that the substrate addition didn't appear to have a huge impact on benthic community structure in the experiments. While these are the results that have been collected and

need to be reported, my feeling is that a lot of the fauna in the substrate addition treatment were actually dead but hadn't decomposed at the end of the experiment. Then, when the fauna were preserved in formalin after 11 days everything that was alive and dead at the end of the experiment was preserved such that no change in community composition could be detected. I understand that this is difficult to assess using staining methods (as stated in the discussion by the author), but it would have been possible to assess the condition of some of the meiofauna at the end of the experiment (e.g., by looking at the appearance of the striated-muscles of the harpacticoids from the burial treatment, and comparing with the control samples). Similar approaches have been undertaken in the past (see Thistle et al. 2005 , Mar. Ecol. Prog. Ser. 289: 1-4) to estimate the proportion of meiofaunal harpacticoids killed in situ by CO2 perturbations. I would suggest that the lack of information about meiofaunal death is clearly flagged as a possible reason why differences in benthic community composition could not be detected. Although the authors went some way to discuss meiofaunal death in their discussion, this point really needs to be stressed. This is because, at present, mining contractors may use this paper to state that manganese nodule particle/ sediment deposition does not alter benthic community composition, and I am not convinced this will be the case. I recommend that the article be published eventually following some moderate revisions. Minor points to consider:

Authors reply: We thank the reviewer for this comment and for the suggestions made. We are aware of the studies using body conditions (muscles and internal organs) of harpacticoids and nematodes. However, own experiments have shown that body condition of freshly killed nematodes and those that were dead since the start of the experiment were comparable until 16 days into the experiment. This (unpublished) experiment was done on an intertidal sediment community. We therefore fear that this method of assessment may be unreliable for short-term experiments (less than 2 weeks). The issue of possible mortality is discussed on several occasions but due to the lack of data on this from our experiment, we think that a more extensive discussion of this topic is too hypothetical and that the absence of mortality could be equally true.

Abstract

Reviewers comment: 1) Line 11: change to "may rive the extraction of deep-sea mineral..."

Authors reply: We replaced "may drive the prospection and exploration" with "may drive the prospection and exploitation".

Reviewers comment: 2) Line 13: Change to "Experimental studies are scarce and simulated effect studies are small scale relative to the effects that will be seen during deep-sea mining..."

Authors reply: As our conducted experiment is extremely small-scale, this sentence would not particularly highlight/relate to this study. We would like to keep the original sentence.

Reviewers comment: 3) Line 16: Insert "in 2015" after conducted.

Authors reply: adjusted as suggested

Reviewers comment: 4) Line 22: Remove "original"

Authors reply: adjusted as suggested

Introduction:

Reviewers comment: 1) Page 2, Line 10: It would be good to provide the range of typical manganese nodule growth rates here, because <250mm myr-1 can mean 0.00000000001mm myr-1 to 250mm myr-1.

Authors reply: The sentence was adjusted to read "with very slow formation and growth rates of 5 to 250 mm My-1 (million years) in the Peru Basin (Von Stackelberg, 2000)." The citation of Jain et al, 1999 was removed as it refers to nodules from the Central Indian Ocean and does not provide clear estimates of nodule growth.

Reviewers comment: 2) Page 2, Line 15: What about organic matter dilution as well

[Figure]

Authors reply: We added organic matter dilution and redistribution. The sentence now reads: "... removal of surface sediment, sediment compaction, sediment suspension and deposition, organic matter dilution and redistribution, discharge of waste material..."

Reviewers comment: 3) Page 2, Line 18: change "of" to "from"

Authors reply: adjusted as suggested

Reviewers comment: 4) Page 2, Line 22: It would be good to give the reader some idea about the natural sedimentation rates in the abyss, and some indication of the levels of sedimentation that will occur during deep sea mining.

Authors reply: A sentence was added on Page 2, Line 20 that states "Sedimentation rates in nodule areas are slow and range between 0.2-1.15 cm kyr-1 (Volz et al., 2018) while sediment resuspension resulting from nodule mining may result in sediment resuspension of 0.6 m$^3$ s-1 (Oebius et al., 2001), therefore, greatly exceeding natural sedimentation rates." The citation "Volz, J.B., Mogollón, J.M., Geibert, W., Martinez-Arbizu, P., Koschinsky, A., Kasten, S., 2018. Natural spatial variability of depositional conditions, biogeochemical processes and element fluxes in sediments of the eastern Clarion-Clipperton Zone, Pacific Ocean. Deep Sea Research Part I: Oceanographic Research Papers 140, 159–172. https://doi.org/10.1016/j.dsr.2018.08.006" was added to the reference list.

Reviewers comment: 5) Page 3, line 2: Change to "which causes at worse, meiofaunal death, but at least removal..."

Authors reply: adjusted as suggested

Reviewers comment: 6) Page 3, line 24-26: I am confused as to why the amount of metal in the animal tissues is a robust way to assess toxic effects. You could have an animal with a high level of metals in its tissues, but the animal is highly resilient to metal toxicity. Therefore, the amount of metal in its tissue does not really always show the

degree of toxicity from that particular metal.

Authors reply: We thank the reviewer for this comment and agree. The sentence was rephrased to read: "Therefore, direct measurements of metals in animal tissues may be used to inform about changes in metal uptake induced by polymetallic nodule mining and may indicate physiological responses to increased metal burdens."

Methods:

Reviewers comment: Overall methods. Did you assess the volume of the sediment taken up by solid nodule particles in your 10cm2 sample from the controls and burial treatments. If some sediments have more solid nodule particles, then there is less sediment to inhabit and this may have an effect on the densities that you found.

Authors reply: The average thickness of the crushed nodule layer in all samples ranged between 1.5 and 2.5 cm. The nodule particle mixture was the same in all 3 cores of the Burial treatment and no additional sediment was added. Therefore, we do not believe this to have influenced meiofauna densities.

Reviewers comment: 1) Page 4, line 8: You need to mention how you sampled the nodule and crushed the nodule to make the substrate. This information is missing.

Authors reply: Page 4 Line 15: We added the missing information. "To obtain the crushed nodule particles, several nodules from the experimental site were collected 2 days prior to the experiment. Upon retrieval, epifauna, if present, was removed from the nodules and nodules were thoroughly washed with fresh water to remove all sediment and fauna. Subsequently, nodules were put inside plastic bags and manually crushed with a hammer. The resulting nodule particles varied in size between 3 $\mu$m and 1 cm (Supplementary Figure S2). "

Reviewers comment: 2) Page 5, line 8: Change to "The second push core was used to..."

Authors reply: Sentence changed to "The second push core was used to analyse sediment characteristics . . ."

Reviewers comment: 3) Page 5, line 9: Did you try and get an idea of the organic matter quality of the sediment and the added substrate? Given that a lot of meiofauna directly consume labile microbial organic matter (see Bernhard & Bowser. 1992. Mar. Ecol. Prog. Ser. 83: 263-272, Ingels et al. 2010. Mar. Ecol. Prog. Ser. 406: 121–133.), the quality of the substrate, as well as the effects from burial and the content of the manganese substrate may have all had an influence on the meiofaunal response. If you do not have actual Chl-a, or lipid concentrations, you can at least get an idea from the C: N ratio.

Authors reply: Indeed, we do not have information on chl a and lipid concentration. We added a sentence on C/N ratios, that were rather similar between the crushed nodule particles and sediment of the control. A sentence was added in the results section Page 7 Line 23: "Despite the lower carbon and nitrogen content in the nodule particles, C/N ratio remained similar between the nodule particles (1.926 $\pm$ 0.037) and the Control sediment (1.951 $\pm$ 0.177)."

Reviewers comment: 4) Page 6, line 18: Please define "live time". It sounds cool, but I have no clue what this is.

Authors reply: Live time is the real time corrected for the "dead time" when the detector is processing the data and not measuring any signal. To avoid confusion we have removed the term "live time" as it is not essential for the understanding of this sentence.

Reviewers comment: 5) Page 7, line 1: What Simpson metric are you referring to? The term 'Simpson's' can actually refer to any one of 3 closely related indices (Simpson's Index, Simpson's Index of Diversity or Simpson's Reciprocal Index).

Authors reply: In our case the Simpson's Index of Diversity was used 1-D=1-$(\sum\_iN\_i(N\_i-1))/(N(N-1))$.

We changed "Shannon-Wiener, Pielou's evenness and Simpson)" to "Shannon-Wiener

index using the natural logarithm (H'), Pielou's evenness (J') and Simpson's index of diversity (1-D))".

Reviewers comment: 6) Page 7, line 1: What univariate analyses were used? Authors reply: Univariate measures were tested as described later on Page 7 Line 10-13. Here, we added "diversity indices" in "Differences of univariate measures (bulk sediment metal contents, total meiobenthos densities and diversity indices) between treatments were tested with a student's t-test"

Results:

Reviewers comment: 1) Page 10, line 13: I think that the biodiversity metrics being the same in both the burial and control treatments may be due to you not being able to differentiate between live and dead fauna. This could have been assessed in the harpacticoids by looking at the condition of the fauna, since dead fauna would appear more degraded even if they've been at abyssal temperatures for a few days. As I stated before, it is important that the manuscript is carefully worded to reflect this as this result could be used as evidence for no impact from re-sedimentation of sediment and nodule particles during mining, and I doubt this will be the case given the low background sedimentation rates in the abyss.

Authors reply: Please see our reply to comment #1

Reviewers comment: 2) Regarding my first point in the methods section above, it would probably have been a good idea to standardise your meiofauna abundances to per unit volume of sediment rather than area. If the nodule substrate layer was full of cm-sized particles then the amount of living space available to the nematodes would be significantly less than in the control samples. Standardising the abundances to unit volume (if you have the data) may show much larger differences, and you may detect differences in community structure, or abundance (at least) between treatments.

Authors reply: Unfortunately, we do not have measurements on the ratios between

small-sized and large-sized nodule particles. The added substrate was a mixture of very fine to very coarse material and a comparison of "living space" for the meiofauna would be difficult.

Discussion:

Reviewers comment: 1) Page 13, line 19: Given the coarse nature of the nodule particles, wouldn't O2 penetrate through the manganese substrate layer relatively easily. I understand there is burial, but diffusion will be dependent on the porosity, which should be greater in the substrate layer.

Authors reply: We thank the reviewer for this remark. It is very unfortunate that we were nog able to measure oxygen penetration. Since the nodule particle mixture contained coarse and fine material, with different settling velocities, the very fine material likely settled on top of the coarse grains, which could have acted as a "seal". Furthermore, an oxygen consumption of the nodule particles themselves may have reduced oxygen concentrations. However, these are merely hypotheses and could not be verified due to the lack of data.

Reviewers comment: 2) More overall impression of the discussion is that the authors need to acknowledge the weaknesses of the study (e.g., being unable to document meiofauna death) to a much better degree. This is done somewhat, but it really needs to be emphasized that a lot of the responses seen (or lack of them, e.g., in the biodiversity data) may be caused by the inability to distinguish living from dead fauna in the different treatments.

Authors reply: Again, we would like to refer to our reply to comment #1 and add that while we share the fear of unnoticed mortality, it would not be correct to emphasize this too much as the opposite "lack of mortality" could be equally true. Nevertheless, we added a sentence op Page 15 Line3 stating "Therefore, potential unnoticed mortality in our study may have masked more severe changes in terms of meiofauna densities and diversity."

---

## Author Comment (AC3) · 7 Mar 2019

General comments:

Reviewers comment: This study explores a really challenging question, that of how deep sea meiofauna respond to mining operations. It is an increasingly vital question as we learn more about the diversity and importance of deep sea meiofauna and as deep sea mining operations expand. I applaud the authors efforts to tackle this problem and I think this study should be published but with some clarification and moderate revisions. My biggest issues with the article center around their methodology and interpretation of depositing nodule sediment onto existing sediment. First, there is no

indication that I can see of where this nodule sediment came from? How far from the "regular" sediment on which it was deposited was it collected? Are these nearby habitats or hundreds of miles apart? Also, why did the authors choose to deposit nodule sediment alone when in their description of nodule mining practices it seems that there is removal of nodule sediment, disturbance of underlying or neighboring sediment, and deposition of nodule particles mixed with suspended sediment and redeposited. It seems the mining operations are after the nodule sediment in particular, so why would they ever redeposit it onto non-nodule sediment? Unless by accident? Please clarify where the nodule sediment came from and why its direct deposition onto non-nodule sediment was chosen as the primary methodology as this doesn't seem to mimic any aspect of the mining operations under question. Also, the authors mention (with citations) in the discussion that meiofauna does inhabit the nodule sediment, yet there seems to be no taking this into account when interpreting the behavior of the meiofauna upon burial. Was the nodule sediment sterilized? Was it presumed that the meiofauna washed out upon transport? It seems like the primary interpretation of the presence of meiofauna in the nodule sediment at the end of the study is that it was colonized from the buried sediment below due to upward movement, but couldn't there have been a meiofaunal community in the nodule sediment upon deposition? If you didn't remove the meiofauna or examine it beforehand, how do you know that meiofauna found in it afterward came from the buried sediment?

Authors reply: For the first remark (origin of the nodule particles), we have added a short paragraph in the Material and Method part on Page 4 Line 15: "To obtain the crushed nodule particles, several nodules from the experimental site were collected 2 days prior to the experiment. Upon retrieval, epifauna, if present, was removed from the nodules and nodules were thoroughly washed with fresh water to remove all sediment and fauna. Subsequently, nodules were put inside plastic bags and manually crushed with a hammer. The resulting nodule particles varied in size between 3 $\mu$m and 1 cm (Supplementary Figure S2). " Thus, the nodule particles originated from the same area and were treated on board prior to the use in the experiments. Because of the

treatment on board (removal of sediment, keeping the particles in plastic bags without the addition of seawater) we are very sure to not have added any meiofauna to the sediment of our experiment. And if meiofauna was present inside the nodule crevices, we would have been able to distinguish them as their shape would appear damaged or dried out from the treatment prior to the experiment. The choice to use crushed nodule particles was partly determined by practical limitations of the experimental design but also to be able to clearly distinguish impacts from the nodule particles with their specific properties (different grain size and porosity, metal content) from the effects of sediment deposition. Especially with regard to metal uptake it was important to limit the study to one substrate instead of a mixture. But indeed, we agree with the reviewer that in a mining scenario, mixtures of sediment and nodule particles will be much more likely. To elucidate the potential source of nodule particles during mining operations a sentence was added in the Introduction on Page 2, Line 13: "Therefore, breakage and abrasion of nodule particles is likely to occur during a mining operation with heavy gear, for example during separation of nodules and sediment as part of the collection process or by the force of the water jet used for the collection of nodules."

Specific comments:

Reviewers comment: Page 2, line 1: "70s" should be "1970's"

Authors reply: adjusted as suggested

Reviewers comment: Page 2, line 17: Here is where you describe mining operations and what happens to nodule sediment and "regular" sediment. You even mention how a large scale mining operation "is expected to directly impact the nodule associated fauna" so then it seems confusing that you then proceed to assume there is no fauna there until you place it on other sediment in your study.

Authors reply: I believe this question relates to our answer to the first general comment. The added nodule substrate did not contain any undamaged meiofauna anymore. Furthermore, from own (unpublished) data we know that the densities of meiofaunal organisms inhabiting crevices of the nodules from the North East Pacific are very low (ranging from 2 to 31 individuals) and, therefore, constitute only a small fraction ($5 \pm 5$ %) of the densities of meiofauna inside the surrounding sediment.

Reviewers comment: Page 4, line 8: Please specify here where the crushed nodule substrate came from and how/if it was treated.

Authors reply: See reply to first general comment.

Reviewers comment: Page 7, line 1: Citations would be helpful for all of these diversity indices and to specify which Simpson index.

Authors reply: In our case the Simpson's Index of Diversity was used 1-D=1-$(\sum\_iN\_i(N\_i-1))/(N(N-1))$.

We changed "Shannon-Wiener, Pielou's evenness and Simpson)" to "Shannon-Wiener index using the natural logarithm (H'), Pielou's evenness (J') and Simpson's index of diversity (1-D))".

Reviewers comment: Page 13, line 10: Please clarify here why you think that all the meiofauna in the nodule sediment came from the lower layer ("adjusting their vertical position in the sediment").

Authors reply: The added substrate did not contain any meiofauna and most meiofaunal organisms, particularly nematodes, do not actively emerge from the sediment. Therefore, it is most likely that colonization of the new substrate was done from the underlying sediment rather than from the water column. This is also in line with the lower densities seen in the 0-2 cm layer of the underlying sediment suggesting that those organisms migrated into the added substrate.

Reviewers comment: Page 14, line 25: Here you mention a study that showed a decrease in nematode densities "attributed to limited upward migration directly after the disturb (as was seen in our experiment)..." but previously you had indicated that there was considerable vertical migration from the lower sediment. Please clarify this.

Authors reply: This part of the discussion was removed as a response to reviewer #1 who suggested that the two studies should not be directly compared due to their very different experimental approach.

Reviewers comment: Page 15, line 27: Here you indicate that your study found that the addition of crushed nodule substrate "changed the relative abundance of feeding types in the new surface layer..." yet you don't seem to have examined the nematodes in the surface layer (nodules) before depositing it, so how can you know this?

Authors reply: Also this comment relates to our reply to the first, general comment. We do believe that the added substrate was void of meiofauna or that meiofauna would at least be very damaged due to the treatment of the nodules prior to the experiment.

─────────────────────────────

---

## Author Response (AR2)

Dear Dr. Treude,

We were very pleased with the suggestion of minor revisions by the reviewers and have taken all of the reviewers' comments into consideration. The manuscript was adjusted according to most suggestions made by both reviewers and we believe that this helped to improve our manuscript substantially. Below we provide a point-by-point answer to all comments raised by the reviewers and indicate how changes were implemented in the manuscript.

We would like to thank you very much in advance for re-considering our manuscript for publication in Biogeosciences.

We are looking forward to your opinion on the revised manuscript.

Sincerely,

On behalf of all co-authors,

Lisa Mevenkamp

**Answers to referee #1: Dr. Rahul Sharma**

We would like to thank Dr. Sharma for his comments on the carried out experiment and for his suggestions to improve the manuscript. We have thoroughly considered all comments and provide our answers and changes to the manuscript below:

**Comment 1:** Having carefully seen the comments of all the reviewers, one thing is common that the 'source' of nodule particles being deposited in such concentrations over a restricted (enclosed) area in real life mining conditions in unclear (and so is the purpose of this experiment). As all the reviewers have pointed out and it is known from literature, that abraded nodule particles dispersed in the sediment plume could settle over a large area and get diluted due to mixing with sediment particles,
10 as against the procedure used in this experiment.

**Reply**: We do agree with the reviewer. This study is done in an experimental setting with an extreme treatment and on a temporal and spatial scale that does not represent a real mining scenario. But, this type of experimental work can shed some light on the responses of meiofauna to a single factor (heavy metal loaded crushed nodule particles) and we think that this is
15 useful as well.
We do not want the reader to believe that this is a representation of responses to a real mining scenario. Therefore, we have modified some sentences in the conclusion to clarify that conditions under a mining scenario are likely to be very different.
Page 17, Line 23-26: "The brittle character of polymetallic nodules implies that deposited material following mining is likely to be a mixture of natural sediment and nodule particles with much lower nodule particle densities. Therefore, our results
20 cannot be directly transferred to such a scenario. Nevertheless; in this research, we revealed some important insights in the structuring of meiobenthic communities following short-term burial with a relatively thick layer of crushed nodule particles."

**Comment 2:** From the description of the experiment, it is clear that it was conducted over a limited (enclosed) area for a short duration (11 days) and to my mind the results have shown positive impacts in terms of no metal accumulation, upward
25 movement of meiofauna-instead of mortality due to smothering (unless there is an issue of preservation/sampling as suggested by one of the reviewers).
**Reply**: We somewhat disagree with the referee here. We do not see reasons to believe that the responses of the meiobenthos are particularly positive. The upward migration is not seen by the animals in the deeper sediment layers and on the longer term, the nodule particle material is not favourable for the migrated fauna due to the very low organic matter content (this may
30 indeed be different in a mining scenario where mixing with the sediment occurs and organic matter gets resuspended).
On the other hand we also don't think that the responses are particularly negative, at least not from what we could find out in our short-term and small-scale experiment. Nevertheless, some aspects, such as unnoticed mortality needed a brief discussion. Overall, we believe that we have sufficiently adjusted the manuscript to not bias the reader in any direction.

35 **Comment 3**: It is well known that any human interference into natural processes would lead to some impact. However, it is not necessary to approach every impact assessment study with a fixed mind of proving that all responses 'have to be' negative. There is always a difference between short term and long term response wherein positive or neutral response is possible and needs to be acknowledged by the scientists.
**Reply**: We agree with the referee, and we can assure that the study was not carried out to prove any negative effects, but to
40 get an idea of the short-term responses of the abyssal fauna to burial. With this mindset we also tried to report our results as they are.

**Comment 4:** Since this experiment is planned as a short term one, results should be expressed as such, which in itself is a contribution to understanding the response of nature to any disturbance. Merely reporting that the impacts of burial are limited
45 (without reporting the actual response) and that long term studies are required, nothing new is being contributed through this experiment and paper.

**Reply**: We believe that this comment relates to the last sentence in the abstract. We agree with the reviewer and changed the sentence to be more specific about our own experiment and its contribution to deep-sea science.

The sentence now reads as, Page 1 Line 28-32: "Our results indicate that burial with a 2 cm layer of crushed nodule particles induces changes in the vertical structure of meiobenthos inside the sediment and an alteration of nematode feeding type proportions on a short time frame of 11 days, while nematode tissue copper burden remains unchanged. These findings considerably contribute to the understanding of the short-term responses of meiobenthos to physical disturbances in the deep sea."

**Comment 5**: I still do not agree with the concept of addressing newly introduced material as 'substrate'. However, I leave it to the authors to provide proper explanation if they want to keep it.

**Reply:** We have replaced "substrate" with "nodule particles", "nodule material" or "crushed nodule particles" where appropriate throughout the manuscript.

Suggestion: The experiment conducted by the researchers is a new approach to the possible impacts of mining of deep-sea minerals and the results should be published. I suggest that the authors should highlight this new approach and report the results 'as they are', because this has not been done before.

Recommendation: The comments above are not meant to discourage the authors but to have a fresh look at the results. The paper may be published after addressing the comments above

**Answers to referee #2**

We would like to thank referee #2 for his/her kind provision of these detailed and helpful suggestions. We have changed the manuscript accordingly and adopted most suggestions. Below we provide our replies to those comments that require an answer or where changes were not adopted as suggested.

**Referee suggestion:** Page 2, Line 10: You may want to say that nodules in the Peru Basin are not commercially exploitable.

**Reply**: We thank the referee for this suggestion, however, we feel that this statement would confuse the reader at this point. The mentioned characteristics (high metal content, slow growth, high porosity) also apply to commercially mineable nodules in the North Pacific (CCZ). To not confuse the reader, we deleted "Peru Basin" and added the reference of Hein et al. 2013, which also includes characteristics of nodules from other Ocean Basins including the CCZ.

**Referee suggestion:** Page 2, line 11: Do you mean shear stress instead of force?

**Reply**: Indeed, shear stress seems to be a better wording.

**Referee suggestion:** Page 2, line 24: Why not cite Alenyik et al. (2017) here who modelled sedimentation rates? You're shifting between resuspension and sedimentation that may be confusing to the reader.

**Reply**: We thank the referee for this suggestion. We changed the sentence to read: "Sedimentation rates in nodule areas are slow and range between 0.2-1.15 cm kyr$^{-1}$ (Volz et al., 2018) while modelled sedimentation rates for a nodule mining scenario in the North Pacific were more than a thousand times higher than the natural background level (Aleynik et al., 2017)."

**Referee suggestion:** Page 6, line 24: MilliQ water?

**Reply**: Changed to "ultrapure water"

**Referee suggestion:** Page 8, Line 8. Change to "treatment"
**Reply**: In this case, the sediment layers in both treatments are discussed. To clarify the sentence we changed "samples" to
5 "sediment layers of both treatments"

**Referee suggestion:** Page 8, Line 11: Change to "…the C/N ratio remained similar between the nodule sediment substrate".
**Reply:** Changed to "the C/N ratio remained similar between the crushed nodule material"

10 **Referee suggestion:** Page 12, Line 10: Please describe which clusters these reductions were seen in.
**Reply**: We added ", in cluster A compared to cluster B"

**Referee suggestion:** Page 16, Line 22: In terms of communities, by specific about what changed. Was it diversity, abundance?
**Reply**: The next sentence in the manuscript explains the community parameters that changed: "…found that nematode density,
15 diversity and community structure inside the disturbed track still differed from adjacent non-disturbed areas." Thus, we do not think that this needs to be also added in the sentence above.

**Referee suggestion:** Page 13, Line 9-10: Could the lack of a difference in the copper burden in the nodule treatment be due to the animals in the nodule substrate treatment being dead and just not taking up copper? This should be discussed.
20 **Referee suggestion:** Page 17, Line 4: I think you need to discuss a little the possible effect of nematode death on the copper burden responses. If the nematodes were all dead in the nodule substrate treatment, then there would be no uptake of copper into their tissues and this may have been the reason for the lack of a difference between the nodule and control treatments. You should discuss this a little.
**Reply**: We thank the reviewer for this remark and suggestion. Indeed, a reduced uptake due to mortality is plausible, however
25 quite speculative. We do not know the exact pathways of the heavy metal uptake in marine nematodes yet, so it would be wrong to draw any conclusions about the nematode state from these measurements. Furthermore, since we do not have any indication of nematode mortality in our experiments, it would be misleading to overemphasize this point in the discussion. We would therefore like to keep it as it is.

Other point that have been changed as suggested:

Page 1, line 16: Change to "at the seafloor of the Peru Basin"
Page 2, line 14: Insert "and transport of nodules"
35 Page 3, line 2: change to "characterized"
Page 3, line 14: Change to "larger organism"
Page 3, line 17: Change to "what thickness"
Page 3, line 18: Change to "evokes meiofaunal responses in the deep sea, and…"
Page 3, line 19: Change to "over short-term timescales".
40 Page 3, line 32: delete comma after depth.
Page 3, line 34: Change to "as well as their vertical structuring in the sediments after eleven days of incubation were assessed"
Page 4, line 4: deep sea, not deep-sea.
Page 4, line 12: Change to ", and these served as experimental controls"
Page 4, line 18: I would remove lines 18-21 and place them after "(Figure 1C)…" in line 14.
45 Page 6, Line 1: Change to "The meiobenthos were…"
Page 6, line 4: Change to "…of De Grisse (1969)"
Page 6, Line 13: Change to "…was carried out"

Page 6, Line 15: Elemental analyser

Page 6, line 23:Change to "…sample taken from" and then delete "taken" in line 24.

Page 7, Line 9: Change to "…with each other. In"

Page 7, Line 18: Change to "…abundance only."

Page 7, line 23: Change to" Visual interpretation of the results was carried out using multidimensional…"

Page 7, Line 28: Change to "…with a non-parametric Wilcox test if datasets failed parametric assumptions of normality and homogeneity of variances…"

Page 10, line 2: Change to "…a similarity level of 92%"

Page 11, Line 8: Change to "…The dominant genera"

Page 11, line 9: Change to "Of the other…"

Page 11, line 14: Change to "branching at a similarity level of 36%"

Page 12, Line 3: Delete commas on either side of "thereby"

Page 15, Line 6: Add a comma after reference.

Page 15, Line 13: Was the reduced C-uptake following 0.5cm or 0.1cm of deposition? Functioning reduced at 0.1, nematode uptake at 05.

Page 15, Line 14: Change to "…reported"

Page 15, Line 18: Change to non-natural substrate.

Page 15, Line 21: Change context to scenarios.

Page 15, Line 27: Change to "…diversity in our treatments".

Page 16, Line 1: Change to "assemblages to be…"

Page 16, Line 12: Change to "abyssal deep sea"

Page 16, Line 17: Add comma after closed bracket of reference.

Page 16, Line 22: Change to "A study by…"

Page 16, Line 22: Change to "…strong but small-scale.."

Page 16, Line 27: Change to "underlie".

Page 17, Line 7: Change to "…levels of trace metals in the porewater returned…"

Page 17, Line 10: Add comma after "processes".

Page 17, Line 12: Change to "contents".

Page 17, Line 13: Change to "mining-related alterations".

Page 17, Line 15: Change to "…implies that deposited.."

Page 17, line 21-22: You didn't vary thicknesses of the nodule layer so I would take this out.

Page 17, line 25: Change to "…was not detected".

Page 17, line 29: Change to "…assessments of fauna in.."

Page 17, Line 30: Change to "…on the benthos".

**Responses of an abyssal meiobenthic community to short-term burial with crushed nodule particles in the South-East Pacific**

Lisa Mevenkamp[1], Katja Guilini[1], Antje Boetius[2], Johan De Grave[3], Brecht Laforce[4], Dimitri Vandenberghe[3], Laszlo Vincze[4], Ann Vanreusel[1]

[1] Department of Biology, Marine Biology Research Group, Ghent University, Ghent, Belgium

[2] HGF MPG Joint Research Group for Deep-Sea Ecology and Technology, Max Planck Institute for Marine Microbiology, Celsiusstr. 1, Bremen, Germany

[3] Department of Geology, Mineralogy and Petrology Research Unit, Ghent University, Ghent, Belgium

[4] Department of Chemistry, X-ray Imaging and Microspectroscopy Research Group, Ghent University, Ghent, Belgium

*Correspondence to*: Lisa Mevenkamp (Lisa.Mevenkamp@ugent.be)

**Abstract.** Increasing industrial metal demands due to rapid technological developments may drive the prospection and exploitation of deep-sea mineral resources such as polymetallic nodules. To date, the potential environmental consequences of mining operations in the remote deep sea are poorly known. Experimental studies are scarce, especially with regard to the effect of sediment and nodule debris depositions as a consequence of seabed mining. To elucidate the potential effects of the deposition of crushed polymetallic nodule particles on abyssal meiobenthos communities, a short (11 days) *in situ* experiment at the seafloor of the Peru Basin in the South East Pacific Ocean was conducted in 2015. We covered abyssal, soft sediment with approx. 2 cm of crushed nodule particles and sampled the sediment after eleven days of incubation at 4200 m water depth. Short-term ecological effects on the meiobenthos community were studied including changes in their composition and vertical distribution in the sediment as well as nematode genus composition. Additionally, copper burden in a few similar-sized, but randomly selected nematodes was measured by means of μ-X-ray fluorescence. At the end of the experiment, 46 ± 1 % of the total meiobenthos occurred in the added crushed nodule layer while abundances decreased in the underlying 2 cm compared to the same depth-interval in undisturbed sediments. Densities and community composition in the deeper 2-5 cm layers remained similar in covered and uncovered sediments. The migratory response into the added substrate nodule material was particularly seen in polychaetes (73 ± 14 %, relative abundance across all depth layers) copepods (71 ± 6 %), nauplii (61 ± 9 %) and nematodes (43 ± 1 %). 
[revised manuscript text omitted]

---

## Author Response (AR3)

Dear Treude,

Thank you for accepting our manuscript for publication in Biogeosciences.

We have implemented the suggested corrections.

Sincerely,

On behalf of all co-authors,

Lisa Mevenkamp

Changes:

Page 7, Line 6: Weblink was removed

Page 18, Line 18: Sentence changed to: "...and nodule particles with much lower nodule particle densities than applied in the present study."

- Page 18, Line 23: "such as bathyal fjords and estuaries" was added.

[revised manuscript text omitted]